# Contrastive Search Is What You Need For Neural Text Generation

**Yixuan Su**                                                                                   *ys484@cam.ac.uk*
*Language Technology Lab, University of Cambridge*

**Nigel Collier**                                                                               *nhc30@cam.ac.uk*
*Language Technology Lab, University of Cambridge*

**Reviewed on OpenReview:** *https://openreview.net/forum?id=GbkWw3jwL9*

## Abstract

Generating text with autoregressive language models (LMs) is of great importance to many natural language processing (NLP) applications. Previous solutions for this task often produce text that contains degenerative expressions (Welleck et al., 2020) or lacks semantic consistency (Basu et al., 2021). Recently, Su et al. (2022b) introduced a new decoding method, *contrastive search*, based on the isotropic representation space of the language model and obtained new state of the art on various benchmarks. In addition, Su et al. (2022b) argued that the representations of autoregressive LMs (e.g. GPT-2) are intrinsically anisotropic which is also shared by previous studies (Ethayarajh, 2019). Therefore, to ensure the language model follows an isotropic distribution, Su et al. (2022b) proposed a contrastive learning scheme, i.e. *SimCTG*, which calibrates the language model's representations through additional training.

In this study, we first answer the question: *"Are autoregressive LMs really anisotropic?"*. To this end, we extensively evaluate the isotropy of LMs across 16 languages. Surprisingly, we find that the anisotropic problem *only* exists in the two specific English GPT-2-small/medium models. On the other hand, *all* other evaluated LMs are isotropic which is in contrast to the conclusion drawn by previous studies (Ethayarajh, 2019; Su et al., 2022b). Based on our findings, we further assess the contrastive search decoding method using *off-the-shelf* LMs on four generation tasks across 16 languages. Our experimental results demonstrate that contrastive search significantly outperforms previous decoding methods *without* any additional training. More notably, on 12 out of the 16 evaluated languages, contrastive search performs comparably with human-level performances as judged by human evaluations. Our code and other related resources are publicly available at https://github.com/yxuansu/Contrastive_Search_Is_What_You_Need.

## 1 Introduction

Natural language generation (NLG) with autoregressive language models (LMs) is an indispensable component of various NLP applications. Some typical examples are dialogue systems (Thoppilan et al., 2022; Rae et al., 2021), contextual text completion (Radford et al., 2019), story generation (Su et al., 2022a), machine translation (NLLB Team et al., 2022), and etc. Predominant approaches for generating text from LMs can be categorized into two classes: (i) maximization-based methods, such as greedy search and beam search; and (ii) stochastic methods, such as top-$k$ sampling (Fan et al., 2018) and nucleus sampling (Holtzman et al., 2020). However, on the one hand, maximization-based approaches tend to produce text that contains undesirable repetitions (Fan et al., 2018; Holtzman et al., 2020) as shown in Table 1. On the other hand,

| Method | Generated Result |
|---|---|
| beam | **Kobe Bryant is** one of the greatest basketball players of all time, but he's also one of the most polarizing figures in the NBA. |
| | Bryant is one of the greatest basketball players of all time, but he's also one of the most polarizing figures in the NBA. |
| | Bryant is one of the greatest basketball players of all time, but he's also one of the most polarizing figures in the NBA. |
| | Bryant is one of the greatest basketball players of all time, but he's also one of the most polarizing figures in the NBA. |
| | Bryant is one of the greatest basketball players of all time, but he's also one of the most polarizing figures in the NBA... |
| nucleus | **Kobe Bryant is** a Hall of Fame Player. Superstar NBA player Steph Curry also is. Scoring a crazy 96 points per game, staving off injury, and making 28 consecutive free throws, Michael Jordan is a cool Hall of Fame Hall of Famer. People may forget he played for a winning team. But remember, basketball Hall of Famers Tom Chambers, Bob Pettit and Walter Davis, along with Pats linebacker Ty Law, played on the losing Dallas Cowboys teams of the 1950's. They played in indestructible stinkin' teams. They were far from perfect. We all remember it. So, if you haven't followed their run to three rings as one of the best defenses in league history, they are hard to forget. There are options available. Jimmy Butler, Dwight Howard, Russell Westbrook, Deron Williams, Ryan Anderson, Patrick Beverley, Anthony Davis, Reggie Jackson, Tony Parker, Kyrie Irving, Gordon Hayward, LeBron James and Damian Lillard are all right now making the All--Defensive teams and likely to be Pro-Bowlers if they stay on this level this season. Shaquille O'Neal is not far behind... |
| contrastive | **Kobe Bryant is** the best player in the world. |
| | I know this is a bold statement to make, but it's true. He may have won a lot of games, but his impact on the game is immeasurable. Kobe is one of the most popular players in the world, and that's no small feat when you consider how many people play video games, watch sports, listen to music, and have social media accounts. |
| | Kobe has made a career out of being a role model for young people, and he does an amazing job of that every time he steps on the court. His career stats are impressive, but what makes him stand out is the way he goes about his business. When he's on the court, he's the center of attention, and it's easy to see why. |
| | The Lakers' All-Star point guard wears a smile on his face like no other. He's always in the right place at the right time, making the right play at the right time, and doing the right thing in the right way. He's not afraid to get his hands dirty, and he never shies away from a physical confrontation... |

Table 1: Texts generated by the *off-the-shelf* GPT-2-large using different methods given the prefix *"Kobe Bryant is"*. (i) Beam search ($b = 5$) generates text with undesirable repetitions (highlighted in red); (ii) Nucleus sampling ($p = 0.95$) quickly goes off-the-topic and talks about other players who are inconsistent with the prefix (highlighted in blue); (iii) Lastly, the text generated by contrastive search is semantically coherent to the prefix while being grammatically fluent. (best viewed in color)

stochastic methods are likely to produce text that is semantically inconsistent with the given human-written prefix (Basu et al., 2021; Su et al., 2022b) (see an example in Table 1).

To address the issues posed by previous studies, Su et al. (2022b) introduced a new decoding method, *contrastive search*, which generates semantically coherent text while maintaining a high level of diversity, based on the isotropic representation space of LMs. Moreover, as widely discussed by previous studies (Ethayarajh, 2019), Su et al. (2022b) argued that autoregressive LMs (e.g. GPT-2) are naturally *anisotropic*, i.e. their token representations reside in a narrow subset of the entire space (Ethayarajh, 2019). Therefore, an additional training stage, i.e. *SimCTG*, is required to calibrate the representation space of LMs. However, an obvious downside of Su et al. (2022b) is that, for extremely large LMs (e.g. GPT-3 (Brown et al., 2020)), this additional training stage is computationally prohibitive which greatly limits the practical applicability of their approach.

While the anisotropy of autoregressive LMs have been widely discussed by previous studies (Ethayarajh, 2019; Su et al., 2022b), in this work, we revisit this problem and try to answer the question: *"Are autoregressive LMs really anisotropic?"*. To this end, we extensively evaluate 38 *off-the-shelf* LMs, ranging from 117M to 30B parameters, across 16 major languages. Surprisingly, we find that the anisotropic problem *only* exists in the two specific English GPT-2-small/medium models. And the rest of evaluated LMs are isotropic which is in contrast to the conclusion drawn by previous studies (Ethayarajh, 2019; Su et al., 2022b).

Based on our findings, we further assess the contrastive search decoding method using *off-the-shelf* LMs on four generation tasks across 16 languages (§4 to §7). Both human and automatic evaluations verify that, *without* any additional training, contrastive search significantly outperforms existing decoding methods and

generates exceptionally high-quality text as shown in Table 1. Furthermore, we provide in-depth analyses on the inner workings of contrastive search (§8).

In summary, our contributions are:

- To the best of our knowledge, our work is the first effort that sheds light on the isotropic property of autoregressive LMs.

- We extensively evaluate contrastive search using *off-the-shelf* LMs from 16 languages across four generation tasks, including (i) open-ended text generation; (ii) document summarization; (iii) code generation; and (iv) machine translation.

- Our experimental results on both human and automatic evaluations verify the clear superiority of contrastive search over existing decoding methods. Notably, on 12 out of the 16 evaluated languages, contrastive search performs comparably with human-level performances.

## 2 Preliminaries

### 2.1 Measurement for the Isotropy of Language Models

To analyze the isotropy of the language model's representation space, we follow previous studies (Ethayarajh, 2019; Su et al., 2021; 2022b) and define the averaged self-similarity of token representations within a text sequence $\boldsymbol{x}$ as

$$\text{self-similarity}(\theta; \boldsymbol{x}) = \frac{1}{|\boldsymbol{x}| \times (|\boldsymbol{x}| - 1)} \sum_{i=1}^{|\boldsymbol{x}|} \sum_{j=1, j \neq i}^{|\boldsymbol{x}|} \frac{h_{x_i}^\top h_{x_j}}{\|h_{x_i}\| \cdot \|h_{x_j}\|}, \tag{1}$$

where $\boldsymbol{x} = \{x_1, ..., x_{|\boldsymbol{x}|}\}$ is a text sequence with variable length; $h_{x_i}$ and $h_{x_j}$ are the token representations of $x_i$ and $x_j$ produced by the language model $\theta$. Intuitively, a lower self-similarity$(\theta; \boldsymbol{x})$ indicates the representations of distinct tokens are less similar, i.e. more discriminative, to each other.

Furthermore, given a text corpus $\mathcal{D} = \{\boldsymbol{x}_i\}_{i=1}^{|\mathcal{D}|}$, we define the isotropy of the language model $\theta$ as

$$\text{isotropy}(\theta) = 1 - \frac{1}{|\mathcal{D}|} \sum_{\boldsymbol{x} \in \mathcal{D}} \text{self-similarity}(\theta; \boldsymbol{x}). \tag{2}$$

Here, a larger isotropy$(\theta)$ means the representations produced by the language model are more evenly distributed in the representation space, therefore better following an isotropic distribution.

### 2.2 Contrastive Search

As discussed in Section §1, to address the issues posed by existing decoding methods, Su et al. (2022b) introduced a new decoding method, *contrastive search*. Formally, given the prefix context $\boldsymbol{x}_{<t}$, the selection of the output $x_t$ follows

$$x_t = \underset{v \in V^{(k)}}{\arg\max} \left\{ (1 - \alpha) \times \underbrace{p_\theta(v | \boldsymbol{x}_{<t})}_{\text{model confidence}} - \alpha \times \underbrace{(\max\{s(h_v, h_{x_j}) : 1 \leq j \leq t - 1\})}_{\text{degeneration penalty}} \right\}, \tag{3}$$

where $V^{(k)}$ is the set of top-$k$ predictions from the language model's probability distribution $p_\theta(\cdot | \boldsymbol{x}_{<t})$. In Eq. (3), the first term, *model confidence*, is the probability of candidate $v$ predicted by the LMs. The second term, *degeneration penalty*, measures the similarity between the candidate $v$ and the tokens in the previous context $\boldsymbol{x}_{<t}$. And $s(\cdot, \cdot)$ computes the cosine similarity between token representations. More specifically, *degeneration penalty* is defined as the maximum cosine similarity between the representation of the candidate $v$ and that of all tokens in $\boldsymbol{x}_{<t}$. Here, the candidate representation $h_v$ is computed by the LMs given the concatenation of $\boldsymbol{x}_{<t}$ and $v$. Intuitively, a larger degeneration penalty of $v$ means it is more similar to the context, therefore

more likely leading to the undesirable repetitions in the generated output. The hyperparameter $\alpha \in [0, 1]$ regulates the importance of these two components.[1] After the selection of output token $x_t$ based on Eq. (3), the representation $h_{x_t}$ is further used to predict the token at time step t+1. In Section §8, we provide in-depth analyses on the inner relationship between contrastive search and the isotropy of LMs.

## 3  Isotropy of Language Models

In this section, we conduct extensive evaluations on the isotropy of LMs from 16 major languages. The model scale of evaluated LMs ranges from 117M to 30B parameters.[2] In Section §3.1, we first evaluate the English LMs. Then, in Section §3.2, we extend our evaluations to multilingual LMs.

**Evaluation Dataset.** To measure the isotropy of LMs from different languages, we use the WIT dataset (Srinivasan et al., 2021) as our text corpus $\mathcal{D}$ (see Eq. (2)). Specifically, WIT consists of general-domain text collected from Wikipedia across 108 languages. For different LMs, we use the text of WIT from the corresponding language to compute the isotropy.

### 3.1  English Language Models

First, we evaluate the isotropy of English LMs. For a comprehensive evaluation, we consider three families of *off-the-shelf* autoregressive LMs.

- GPT-2 (Radford et al., 2019): We evaluate all publicly available GPT-2 models with different model scales, including small (i.e. 117M), medium (i.e. 345M), large (i.e. 774M), and xl (i.e. 1.6B).
- GPT-Neo (Black et al., 2021): We evaluate all three publicly available GPT-Neo models with the parameter size of 125M, 1.3B, and 2.7B, respectively.
- OPT (Zhang et al., 2022): OPT is recently released by Meta as an open-sourced replication of GPT-3 (Brown et al., 2020). In our experiments, we evaluate the OPT model with up to 30B parameters.

Figure 1 plots the isotropy results of different English LMs. On the one hand, we see that, among *all* evaluated LMs, only the small (i.e. 117M) and medium (i.e. 345M) size GPT-2 display a clear anisotropy (i.e. isotropy($\theta$) $\leq 0.25$). On the other hand, the representation space of *all* other evaluated LMs are remarkably better and isotropic.

It is worth emphasizing that previous studies (Ethayarajh, 2019; Su et al., 2022b), which discuss the anisotropy of English autoregressive LMs, *only focus* on the specific GPT-2-small model (i.e. 117M). And our observation on the anisotropy of GPT-2-small is also inline with previous studies (Ethayarajh, 2019; Su et al., 2022b). However, through extensive evaluations on a wide range of LMs with different scales, we empirically show that the majority of existing English autoregressive LMs are isotropic, and this observation also holds for multilingual LMs (§3.2).[3]

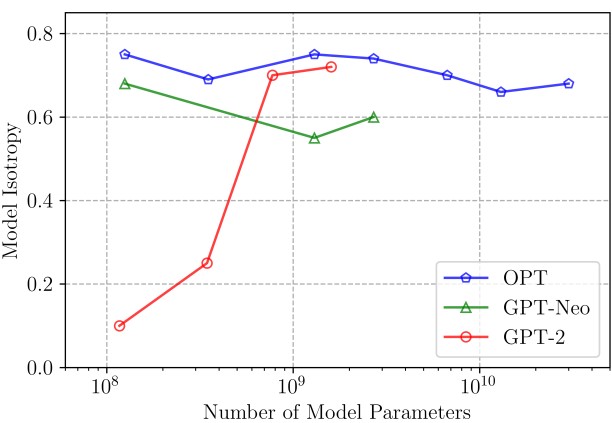

Figure 1: Isotropy results of English LMs.

**Remark.** We acknowledge that there are many factors (e.g. training data, model initialization, optimization, and etc.) that could potentially cause the unusual behaviors of the English GPT-2-small/medium models. The rigorous investigation on these factors is out-of-the-scope of this study and we leave it to our future

---

[1]When $\alpha = 0$, contrastive search degenerates to the greedy search method.
[2]The complete list of evaluated languages as well as LMs is provided in Table 10 at Appendix D.
[3]In Appendix B, we provide more discussions on the isotropy of English GPT-2 models.

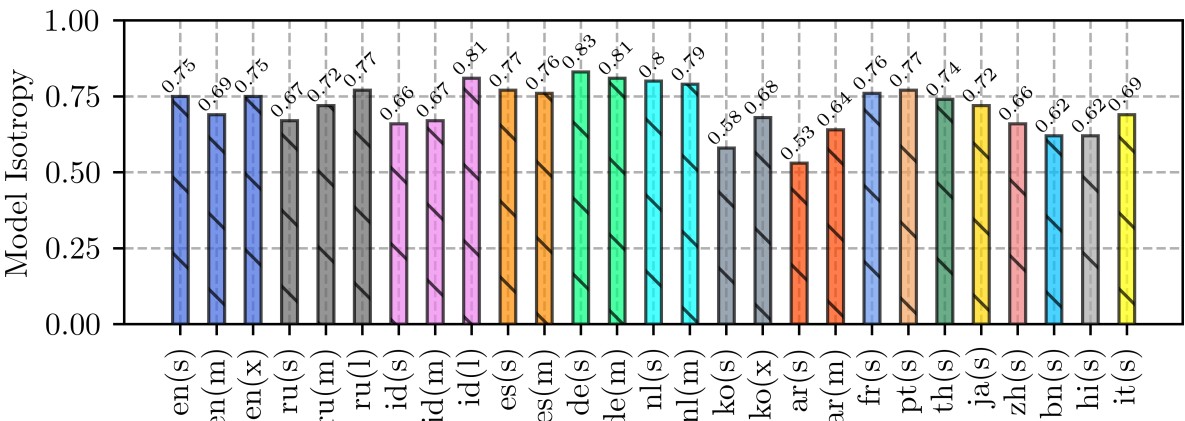

Figure 2: Isotropy results of multilingual LMs. Each x(y) denotes the language code (x) and the model size (y), where s is for small size model (i.e. ∼120M parameters), m is for medium size model (i.e. ∼350M parameters), l is for large size model (i.e. ∼780M parameters), and x is for xl size model (i.e. ∼1.5B parameters). For English (i.e. en) LMs, we plot the results of three OPT models. The detailed list of language codes and evaluated LMs can be found in Table 10 at Appendix D.

work. Nonetheless, based on our extensive evaluations in both English LMs (§3.1) as well as multilingual LMs (detailed in §3.2), we empirically show that the majority of existing autoregressive LMs are isotropic.

### 3.2   Multilingual Language Models

Here, we evaluate the isotropy of multilingual LMs, with different scales, across 16 languages. Figure 2 presents the evaluated results, from which we see that the isotropy scores of *all* evaluated LMs are above 0.50. This clearly indicates that our finding in Section §3.1, i.e. *the majority of existing autoregressive LMs are isotropic*, is generalizable to different languages.

## 4   Open-ended Text Generation

In this section, we present our experimental results on open-ended text generation for both English LMs (§4.1) and multilingual LMs (§4.2). Formally, open-ended generation is defined as, conditioned on a human-written prefix (i.e. context), the language model is required to generate a text continuation that is grammatically fluent while being semantically coherent with the context (Holtzman et al., 2020; Su et al., 2022b; Su & Xu, 2022).

### 4.1   English Open-ended Text Generation

Following previous study (Holtzman et al., 2020), we use the large version of GPT-2 (Radford et al., 2019) (i.e. GPT-2-large) to generate texts conditioned on the initial paragraph (restricted to 40 tokens) of documents from the held-out set of WebText (Radford et al., 2019). Specifically, the generation of text ends upon reaching an end-of-document token or a maximum length of 200 tokens.

#### 4.1.1   Automatic Evaluation

**Decoding Methods.** We compare various decoding strategies, including (1) greedy search; (2) beam search ($b = 4$); (3) typical sampling ($\tau = 0.95$) (Meister et al., 2022); (4) top-$k$ sampling ($k = 50$) (Fan et al., 2018); (5) nucleus sampling ($p = 0.95$) (Holtzman et al., 2020); and (6) contrastive search ($k = 5$, $\alpha = 0.6$) (Su et al., 2022b).[4]

---

[4]The hyperparameters of different methods are selected based on their optimal MAUVE score on the validation set.

**Evaluation Metrics.** Following previous studies (Welleck et al., 2020; Su et al., 2022b), we evaluate the generated results of different decoding methods using (i) **diversity**, which provides an overall assessment on the repetition of generation at different $n$-gram levels and $n \in \{2, 3, 4\}$; and (ii) **MAUVE** (Pillutla et al., 2021), which measures the token distribution closeness between the generated text and the human-written text. A higher MAUVE score means the generated text is more human-like. (iii) Moreover, it has been widely demonstrated that, by simply measuring the log-likelihood of the text, the massively pre-trained language models (Brown et al., 2020; Zhang et al., 2022) display an exceptional zero-shot performance on tasks like sentence completion selection (Zellers et al., 2019) and natural language inference (NLI) (Wang et al., 2018). We follow the same practice and introduce a new **coherence** metric to automatically measure the semantic coherence between the generated text and the given prefix text. Specifically, the metric is defined as the averaged log-likelihood of the generated text conditioned on the prefix text as

$$\text{coherence}(\hat{\boldsymbol{x}}, \boldsymbol{x}) = \frac{1}{|\hat{\boldsymbol{x}}|} \sum_{i=1}^{|\hat{\boldsymbol{x}}|} \log p_{\mathcal{M}}(\hat{\boldsymbol{x}}_i | [\boldsymbol{x} : \hat{\boldsymbol{x}}_{<i}]), \tag{4}$$

where $\boldsymbol{x}$ and $\hat{\boldsymbol{x}}$ are the prefix text and the generated text, respectively; and $[:]$ is the concatenation operation. For the evaluation model $\mathcal{M}$, we choose the recently released OPT (Zhang et al., 2022) which is massively pre-trained on over 180 billion tokens. To alleviate the potential measurement inaccuracy caused by the inductive bias of $\mathcal{M}$, we present the coherence score obtained by OPT with different model scales (i.e. 125M, 2.7B, and 13B parameters, respectively). (iv) Lastly, we also report the averaged length of the generated text (i.e. **gen-length**) from different decoding methods.

| **Method** | diversity(%)↑ | MAUVE(%)↑ | gen-length | coherence↑ | | |
|---|---|---|---|---|---|---|
| | | | | OPT-125M | OPT-2.7B | OPT-13B |
| greedy | 5.38 | 7.91 | 147.28 | $-0.72$ | $-0.58$ | $-0.60$ |
| beam | 4.04 | 5.22 | 137.55 | $\mathbf{-0.59}$ | $\mathbf{-0.46}$ | $\mathbf{-0.46}$ |
| typical | 87.98 (±**0.13**) | 49.76 (±**3.90**) | 142.11 (±**0.70**) | $-2.45$ (±**0.02**) | $-2.20$ (±**0.01**) | $-2.25$ (±**0.01**) |
| top-$k$ | 91.33 (±**0.05**) | **89.64** (±**2.37**) | 142.48 (±**0.28**) | $-2.59$ (±**0.01**) | $-2.35$ (±**0.01**) | $-2.42$ (±**0.01**) |
| nucleus | **93.61** (±**0.07**) | 87.89 (±**0.97**) | 139.49 (±**0.99**) | $-3.09$ (±**0.01**) | $-2.88$ (±**0.01**) | $-2.94$ (±**0.01**) |
| contrastive | 92.54 | 87.26 | 140.72 | $-1.93$ | $-1.52$ | $-1.56$ |

Table 2: Automatic evaluation results on the held-out set of WebText. ↑ means the higher the better.

**Evaluation Results.** Table 2 presents the experimental results.[5] Firstly, as demonstrated by the diversity and MAUVE scores, greedy and beam search stuck in repetitive loops (see an example in Table 1) and produce less human-like results. These repetitions generated by greedy and beam search further lead to the high coherence score (i.e. log-likelihood) as judged by the OPT models.[6] Secondly, we see that contrastive search achieves comparable results with other stochastic methods (i.e. typical, top-$k$, and nucleus sampling) on metrics including diversity, MAUVE, and gen-length. On the other hand, contrastive search performs notably better on the coherence metric as measured by OPT with different scales, suggesting it best maintains the semantic consistency between the generated text and the given prefix text.

### 4.1.2 Human Evaluation

We also conduct a human evaluation with the help of five graders proficient in English from a third-party grading platform. Specifically, we randomly select 200 prefixes from the held-out set of WebText and ask the annotators to assess the generation quality of different decoding methods, including (i) typical sampling; (ii) top-$k$ sampling; (iii) nucleus sampling; and (iv) contrastive search. The evaluation is conducted through pairwise comparisons by jointly considering the following aspects:

- **Coherence**: Whether the generated text is semantically consistent with the prefix text.

---

[5]For stochastic methods, i.e. typical, top-$k$, and nucleus sampling, we report their results averaged over three runs with different random seeds.

[6]Xu et al. (2022) pointed out the *self-reinforcement effect* of autoregressive LMs, i.e. the likelihood of repetition increases with the number of historical repetitions.

- **Fluency**: Whether the generated text is fluent and easy to understand.

- **Informativeness**: Whether the generated text is diverse and contains interesting content.

Table 3 presents the human evaluation results. We can see that contrastive search outperforms all compared baselines by significant margins. It is worth noting that contrastive search even performs comparably with the human-written text as judged by Sign Test. These results indicate that (i) LMs can successfully learn the underlying knowledge (e.g. grammars and linguistic patterns) of human language through large-scale pre-training over unstructured text; and (ii) with the state-of-the-art decoding method, i.e. *contratsive search*, the intrinsic knowledge of LMs can be effectively elicited, therefore producing text with high quality.

| Method A | is Better | Neutral | Method B | is Better |
|---|---|---|---|---|
| contrastive | **71.2**%[†] | 15.3% | 13.5% | typical |
| contrastive | **68.7**%[†] | 14.7% | 16.6% | top-$k$ |
| contrastive | **64.2**%[†] | 15.5% | 20.3% | nucleus |
| contrastive | 19.9%[‖] | 57.9% | **22.2**%[‖] | human |

Table 3: Human evaluation on WebText. [†] means one method performs significantly better than the other as judged by Sign Test with $p$-value $< 0.05$. [‖] means one method performs comparably with the other with $p$-value $> 0.4$.

## 4.2 Multilingual Open-ended Text Generation

Next, we extend our evaluation to multilingual open-ended text generation on 16 languages.

**Evaluation Benchmark.** We conduct experiments on the WIT dataset (Srinivasan et al., 2021) which consists of general-domain text collected from Wikipedia across 108 languages. For each evaluated language, we use the LMs to generate text conditioned on the prefix (restricted to 16 tokens) from the test set of WIT. The generation of text ends upon reaching an end-of-document token or a maximum length of 64 tokens.

**Experiment Setups.** To generate text, we use GPT-2 models from different languages that are publicly available in the Huggingface library (Wolf et al., 2019). We compare the results of contrastive search with the strong baseline, i.e. nucleus sampling ($p = 0.95$).[7] For the assessment of generated results, we rely on human evaluation following the same protocol as described in Section §4.1.2.

**Evaluation Results.** Our experimental results are presented in Table 4. From the results, we see that, for *all* evaluated languages, contrastive search significantly outperforms nucleus sampling as validated by Sign Test. Furthermore, on 12 out of the 16 evaluated languages (i.e. except for *Hindi*, *Thai*, *Indonesia*, and *Russian*), the performances of contrastive search are comparable with human-written texts. These evaluation results clearly demonstrate the generalization ability of contrastive search across different languages as well as its superiority over existing decoding methods.

## 5 Document Summarization

In this section, we present our experimental results on the task of document summarization.

**Benchmark.** We use the widely-used XSum dataset (Narayan et al., 2018) as our test bed which consists of news articles collected from BBC along with the corresponding one-sentence summaries.

**Models and Decoding Methods.** We conduct experiments using OPT models with different scales, ranging from 125M to 2.7B parameters. To generate the summary, we apply different decoding methods, including beam search ($b = 4$), nucleus sampling ($p = 0.95$), and contrastive search ($k = 5, \alpha = 0.6$).

---

[7]The details of (i) evaluated languages; (ii) the link address of assessed LMs; and (ii) the hyperparameters of contrastive search are provided in Table 11 at Appendix E.

| Spanish | | | | French | | | |
|---|---|---|---|---|---|---|---|
| Method A is Better | | Neutral | Method B is Better | Method A is Better | | Neutral | Method B is Better |
| contrastive | **71.2**%[†] | 20.0% | 8.8% | nucleus | contrastive | **86.3**%[†] | 2.7% | 11.0% | nucleus |
| contrastive | 16.3%[‖] | 63.9% | **19.8**%[‖] | human | contrastive | 22.8%[‖] | 53.1% | **24.1**%[‖] | human |

| Chinese | | | | Hindi | | | |
|---|---|---|---|---|---|---|---|
| Method A is Better | | Neutral | Method B is Better | Method A is Better | | Neutral | Method B is Better |
| contrastive | **92.7**%[†] | 4.9% | 2.4% | nucleus | contrastive | **47.9**%[†] | 20.8% | 31.3% | nucleus |
| contrastive | **30.4**%[‖] | 41.8% | 27.8%[‖] | human | contrastive | 20.4% | 43.5% | **36.1**%[†] | human |

| Thai | | | | Indonesia | | | |
|---|---|---|---|---|---|---|---|
| Method A is Better | | Neutral | Method B is Better | Method A is Better | | Neutral | Method B is Better |
| contrastive | **68.1**%[†] | 4.3% | 27.6% | nucleus | contrastive | **65.4**%[†] | 6.0% | 28.6% | nucleus |
| contrastive | 18.9% | 49.4% | **31.7**%[†] | human | contrastive | 16.9% | 44.3% | **38.8**%[†] | human |

| Arabic | | | | Japanese | | | |
|---|---|---|---|---|---|---|---|
| Method A is Better | | Neutral | Method B is Better | Method A is Better | | Neutral | Method B is Better |
| contrastive | **84.1**%[†] | 2.0% | 13.9% | nucleus | contrastive | **62.1**%[†] | 18.0% | 19.9% | nucleus |
| contrastive | **24.6**%[‖] | 52.6% | 22.8%[‖] | human | contrastive | 30.3%[‖] | 33.1% | **36.6**%[‖] | human |

| English | | | | Bengali | | | |
|---|---|---|---|---|---|---|---|
| Method A is Better | | Neutral | Method B is Better | Method A is Better | | Neutral | Method B is Better |
| contrastive | **72.3**%[†] | 15.6% | 12.1% | nucleus | contrastive | **73.7**%[†] | 8.1% | 18.2% | nucleus |
| contrastive | 23.3%[‖] | 51.9% | **24.8**%[‖] | human | contrastive | 24.8%[‖] | 48.6% | **26.6**%[‖] | human |

| Korean | | | | German | | | |
|---|---|---|---|---|---|---|---|
| Method A is Better | | Neutral | Method B is Better | Method A is Better | | Neutral | Method B is Better |
| contrastive | **69.2**%[†] | 12.3% | 18.5% | nucleus | contrastive | **76.8**%[†] | 13.3% | 9.9% | nucleus |
| contrastive | **29.8**%[‖] | 44.6% | 25.6%[‖] | human | contrastive | **30.2**%[‖] | 43.0% | 26.3%[‖] | human |

| Italian | | | | Portuguese | | | |
|---|---|---|---|---|---|---|---|
| Method A is Better | | Neutral | Method B is Better | Method A is Better | | Neutral | Method B is Better |
| contrastive | **69.7**%[†] | 11.9% | 18.4% | nucleus | contrastive | **75.8**%[†] | 13.1% | 11.1% | nucleus |
| contrastive | 23.7%[‖] | 50.2% | **26.1**%[‖] | human | contrastive | **30.2**%[‖] | 43.9% | 25.9%[‖] | human |

| Dutch | | | | Russian | | | |
|---|---|---|---|---|---|---|---|
| Method A is Better | | Neutral | Method B is Better | Method A is Better | | Neutral | Method B is Better |
| contrastive | **85.6**%[†] | 10.2% | 4.2% | nucleus | contrastive | **48.2**%[†] | 21.3% | 30.5% | nucleus |
| contrastive | **33.2**%[‖] | 40.0% | 26.8%[‖] | human | contrastive | 18.9% | 41.3% | **39.8**%[†] | human |

Table 4: Human evaluation on multilingual open-ended text generation. [†] means one method performs significantly better than the other as judged by Sign Test with $p$-value $< 0.05$. [‖] means one method performs comparably with the other with $p$-value $> 0.4$.

**Evaluation Setups.** We test the model under two settings: zero-shot learning and in-context learning. (i) For the zero-shot setting, given the article (e.g. *"This is an article."*), we provide a natural language input *"Article:\n\n This is an article.\n\n Summary:"* to the model, and let it generate the summary autoregressively. (ii) For the in-context learning setting, when generating the summary, we follow previous studies (Brown et al., 2020; Zhang et al., 2022) and additionally provide the model with one or two in-context examples. Here, each in-context example is a pair of article and the corresponding summary.[8]

---

[8]As the articles are generally quite long (i.e. over several hundreds of tokens), we can only provide up to two in-context examples to the LMs.

| Shot | Method | OPT-125M | | | OPT-350M | | | OPT-1.3B | | | OPT-2.7B | | |
|------|--------|------|------|------|------|------|------|------|------|------|------|------|------|
| | | R-1 | R-2 | R-L | R-1 | R-2 | R-L | R-1 | R-2 | R-L | R-1 | R-2 | R-L |
| | beam | 9.05 | 1.24 | 6.78 | 1.46 | 0.23 | 1.11 | 11.35 | 1.51 | 8.50 | 5.76 | 0.85 | 4.36 |
| Zero | nucleus | 10.25 | 0.70 | 7.80 | **5.26** | **0.35** | **4.03** | 12.56 | 1.32 | 9.22 | **6.59** | 0.96 | **4.74** |
| | contrastive | **12.68** | **1.75** | **9.59** | 1.11 | 0.16 | 0.86 | **16.76** | **3.17** | **12.64** | 4.95 | **1.03** | 3.81 |
| | beam | 13.48 | 1.28 | 10.17 | 15.50 | 2.04 | 11.61 | 23.37 | 5.81 | 18.07 | 24.99 | 6.92 | 19.50 |
| One | nucleus | 12.36 | 0.84 | 9.49 | 12.27 | 0.77 | 9.38 | 10.01 | 1.80 | 11.61 | 18.14 | 3.03 | 13.82 |
| | contrastive | **15.86** | **1.96** | **12.03** | **17.30** | **2.67** | **13.27** | **25.36** | **6.57** | **19.76** | **27.77** | **8.22** | **21.77** |
| | beam | 17.02 | 2.02 | 12.97 | 17.66 | 2.30 | 13.58 | 25.36 | 7.10 | 19.72 | 25.85 | 7.72 | 20.42 |
| Two | nucleus | 12.75 | 0.87 | 9.71 | 12.45 | 0.99 | 9.69 | 17.99 | 2.89 | 13.69 | 19.07 | 3.57 | 14.64 |
| | contrastive | **18.04** | **2.63** | **13.89** | **18.84** | **3.01** | **14.48** | **27.31** | **7.54** | **21.16** | **29.02** | **9.09** | **23.07** |

Table 5: Experimental results on the XSum benchmark, in which R-1, R-2, R-L denote ROUGE-1, ROUGE-2, and ROUGE-L (Lin, 2004), respectively.

**Results.** Table 5 presents the evaluation results on XSum.[9] On the one hand, under the zero-shot setting, the performance of different methods are fluctuated across different LMs. We conjecture that such instability comes from the distinct inductive bias of LMs with different scales. On the other hand, by providing one or two in-context examples, we observe much better performances from the LMs which demonstrates its strong in-context learning ability (Brown et al., 2020; Zhang et al., 2022). Moreover, across all evaluation metrics, contrastive search consistently achieves the best results with notable margins, demonstrating its clear advantages over existing decoding methods.

## 6 Code Generation

We also conduct experiments on the task of code generation. In this task, given a natural language prompt, the LMs is required to generate a complete code snippet that fulfills the function specified by the prompt. Following previous studies (Chen et al., 2021; Nijkamp et al., 2022), we use the HumanEval dataset (Chen et al., 2021) as our testbed. We apply the CodeGen model (Nijkamp et al., 2022) with two model scales (i.e. 350M and 2B parameters) and generate the code with three decoding methods, including beam search ($b = 4$), nucleus sampling ($p = 0.95$), and contrastive search ($k = 3, \alpha = 0.4$).

| Model | Method | Pass Rate@1 (%) |
|-------|--------|-----------------|
| | beam | 14.63 |
| CodeGen-350M-mono | nucleus | 5.08 (±**0.76**) |
| | contrastive | **15.24** |
| | beam | 18.90 |
| CodeGen-2B-mono | nucleus | 10.98 (±**0.50**) |
| | contrastive | **21.95** |

Table 6: Code generation results on HumanEval dataset.

The evaluation results[10] on Pass Rate@1 are shown in Table 6, from which we can draw the same conclusion that contrastive search outperforms other decoding methods.

---

[9]For one- and two-shot settings, we report the results of different methods over three random selections of in-context examples. We provide the detailed results in Table 12 at Appendix F.

[10]For the stochastic method, i.e. nucleus sampling, we report the averaged results over three different runs. The detailed results are provided in Table 13 at Appendix G.

# 7 Machine Translation

Lastly, we conduct experiments on the machine translation task using the IWSLT14 De-En dataset. Same as in Section §5, we test OPT models[11] with different scales using three decoding methods: (i) beam search ($b = 4$); (ii) nucleus sampling ($p = 0.95$); and contrastive search ($k = 3, \alpha = 0.4$).

| Shot | Method | OPT-125M | | OPT-350M | | OPT-1.3B | | OPT-2.7B | |
|---|---|---|---|---|---|---|---|---|---|
| | | BLEU | COMET | BLEU | COMET | BLEU | COMET | BLEU | COMET |
| One | beam | $0.00\,(\pm0.00)$ | $-1.24\,(\pm0.09)$ | $\mathbf{0.03}\,(\pm\mathbf{0.04})$ | $-\mathbf{1.26}\,(\pm\mathbf{0.05})$ | $5.53\,(\pm1.33)$ | $-0.55\,(\pm0.14)$ | $\mathbf{14.06}\,(\pm\mathbf{0.67})$ | $\mathbf{0.07}\,(\pm\mathbf{0.05})$ |
| | nucleus | $0.00\,(\pm0.00)$ | $-1.49\,(\pm0.04)$ | $0.00\,(\pm0.00)$ | $-1.54\,(\pm0.02)$ | $2.18\,(\pm0.68)$ | $-0.85\,(\pm0.13)$ | $7.15\,(\pm0.64)$ | $-0.22\,(\pm0.05)$ |
| | contrastive | $\mathbf{0.05}\,(\pm\mathbf{0.07})$ | $-\mathbf{1.18}\,(\pm\mathbf{0.11})$ | $0.00\,(\pm0.00)$ | $-1.30\,(\pm0.01)$ | $\mathbf{7.10}\,(\pm\mathbf{1.33})$ | $-\mathbf{0.41}\,(\pm\mathbf{0.08})$ | $12.98\,(\pm0.77)$ | $0.04\,(\pm0.04)$ |
| Few | beam | $0.00\,(\pm0.00)$ | $-1.45\,(\pm0.09)$ | $\mathbf{0.08}\,(\pm\mathbf{0.11})$ | $-\mathbf{1.40}\,(\pm\mathbf{0.07})$ | $\mathbf{8.54}\,(\pm\mathbf{0.75})$ | $-0.36\,(\pm0.09)$ | $\mathbf{14.59}\,(\pm\mathbf{0.40})$ | $\mathbf{0.08}\,(\pm\mathbf{0.01})$ |
| | nucleus | $0.03\,(\pm0.05)$ | $-1.54\,(\pm0.04)$ | $0.03\,(\pm0.05)$ | $-1.57\,(\pm0.06)$ | $4.22\,(\pm0.68)$ | $-0.54\,(\pm0.06)$ | $8.36\,(\pm0.30)$ | $-0.15\,(\pm0.03)$ |
| | contrastive | $\mathbf{0.05}\,(\pm\mathbf{0.07})$ | $-\mathbf{1.38}\,(\pm\mathbf{0.12})$ | $0.05\,(\pm0.07)$ | $-1.41\,(\pm0.09)$ | $8.39\,(\pm0.71)$ | $-\mathbf{0.29}\,(\pm\mathbf{0.05})$ | $13.52\,(\pm0.14)$ | $0.05\,(\pm0.01)$ |

Table 7: Machine translation results on IWSLT14 De-En dataset.

Table 7 presents the BLEU-4 (Papineni et al., 2002) and COMET (Rei et al., 2020) results under one-shot and few-shot settings.[12] Firstly, we see that smaller LMs (i.e. model scale $\leq$ 350M) does not yield satisfactory BLEU scores. In contrast, by scaling up the model parameters, larger LMs (i.e. model scale $\geq$ 1.3B) starts to display emergent capability (Wei et al., 2022) and obtains notably better results. Secondly, contrastive search consistently outperforms nucleus sampling but performs slightly worse than beam search on a few evaluations on both the BLEU and COMET metrics. This reveals the advantage of maximization-based decoding methods, i.e. beam search, in tasks like machine translation that demand a high surface-level accuracy (e.g. BLEU).

# 8 Further Analysis

## 8.1 Relationship between Contrastive Search and the Isotropy of LMs

We conduct quantitative analysis on the importance of LMs' isotropy for contrastive search.[13] To this end, given a prefix text $\boldsymbol{x}$, we measure the variance of degeneration penalty (see Eq. (3)) as

$$\mathrm{dp}(v; \boldsymbol{x}, \theta) = \max\{s(h_v, h_{x_j}) : 1 \leq j \leq |\boldsymbol{x}|\},$$
$$\mathrm{var}(\boldsymbol{x}; \theta) = \sqrt{\frac{1}{k} \sum_{v \in V^{(k)}} (\mathrm{dp}(v; \boldsymbol{x}, \theta) - \mu)^2}, \tag{5}$$

where $s(\cdot, \cdot)$ computes the cosine similarity between token representations; $V^{(k)}$ is the set of top-$k$ predictions from the language model's probability distribution $p_\theta(\cdot|\boldsymbol{x})$; and $\mu = \frac{1}{k} \sum_{v \in V^{(k)}} \mathrm{dp}(v; \boldsymbol{x}, \theta)$. Then, we define the averaged variance of degeneration penalty at each decoding step $t$ as

$$f(t; \theta, \mathcal{D}) = \frac{1}{\mathcal{D}} \sum_{\boldsymbol{x} \in \mathcal{D}} \mathrm{var}([\boldsymbol{x} : \hat{\boldsymbol{x}}]; \theta), \tag{6}$$

where $\mathcal{D}$ is a text corpus; $\boldsymbol{x}$ is the prefix text with a fixed length; $\hat{\boldsymbol{x}}$ is the text continuation generated by $\theta$ using contrastive search and $|\hat{\boldsymbol{x}}| = t$.

In our experiments, we follow similar procedures as in Section §4.1. Specifically, we use GPT-2 models with different scales to generate text (up to 200 tokens) conditioned on the initial paragraph (restricted to 40

---

[11]While our experiments in this study primarily focus on autoregressive LMs, in Appendix I, we provide more experimental results with encoder-decoder models.

[12]Under few-shot setting, 8 in-context examples are provided to the LMs. We report the results averaged over three random selections of in-context examples. The detailed results are presented in Table 14 at Appendix H.

[13]Su et al. (2022b) only qualitatively pointed out that, to apply contrastive search, the representation space of the LMs should be isotropic.

tokens) of documents from the held-out set of WebText. The $k$ and $\alpha$ in contrastive search are set as 5 and 0.6, respectively.

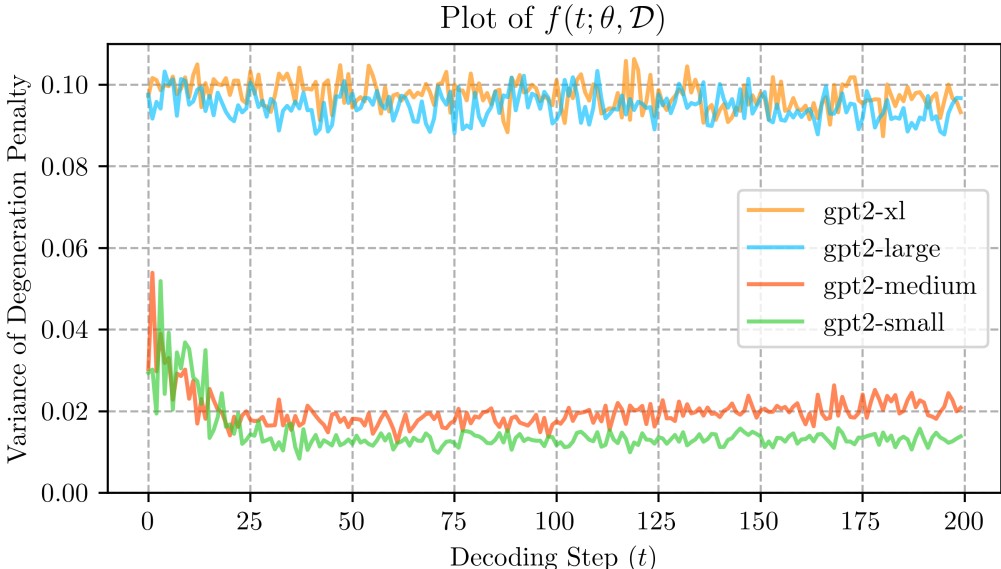

Figure 3: Averaged variance of degeneration penalty of different GPT-2 models.

Figure 3 plots the results of different GPT-2 models[14] over the decoding steps. On the one hand, for GPT-2-small/medium models that have a low isotropy in the representation space (§3.1), their averaged variance of degeneration penalty across the decoding process is closer to 0. In other words, when applying contrastive search (see Eq. (3)), the degeneration penalties of different candidates are indistinguishable to each other. Therefore, the selection of the output is dominated by the model confidence term, making contrastive search degenerate to the greedy search method. On the other hand, with an isotropic representation space (§3.1), the GPT-2-large/xl models display notably higher averaged variance in their degeneration penalties. During the decoding process, such high variance helps the LMs to avoid model degeneration, therefore generating high-quality text. In conclusion, an isotropic representation space of the LMs is essential for contrastive search to work well.

## 8.2 Contrastive Search versus Sampling Methods

Here, we provide further comparisons between contrastive search and other strong sampling methods (i.e. top-$k$ sampling and nucleus sampling). To this end, we follow Section §4.1 and generate text using GPT-2-large with different decoding methods. Specifically, we vary the hyperparameters of different methods, i.e. $k$ for top-$k$ sampling (from 5 to 640); $p$ for nucleus sampling (from 0.4 to 1.0); and $k$ for contrastive search (from 2 to 10).[15]

The generated texts are evaluated from two aspects: (i) MAUVE and (ii) coherence (obtained with the OPT-2.7B model) that are described in Section §4.1.1. Figure 4 plots the results of different decoding methods. We can see that, by varying the hyperparameters, the performances of sampling methods change drastically on the coherence metric. On the other hand, contrastive search best balances the trade-off between MAUVE and coherence. These results further verify the strong robustness of contrastive search over different selections of hyperparameters.[16]

---

[14]The experimental results on other LMs (i.e. GPT-Neo and OPT) are provided in Appendix K.

[15](i) For top-$k$ sampling, $k \in [5, 10, 20, 40, 50, 80, 160, 320, 640]$; (ii) for nucleus sampling, $p \in [0.4, 0.5, 0.6, 0.7, 0.8, 0.9, 0.95, 1.0]$; and (iii) for contrastive search, $k \in [2, 3, 4, 5, 6, 7, 8, 9, 10]$ and we keep $\alpha$ as a constant 0.6.

[16]In Appendix J, we provided detailed ablation studies on the effect of both $k$ and $\alpha$ in contrastive search.

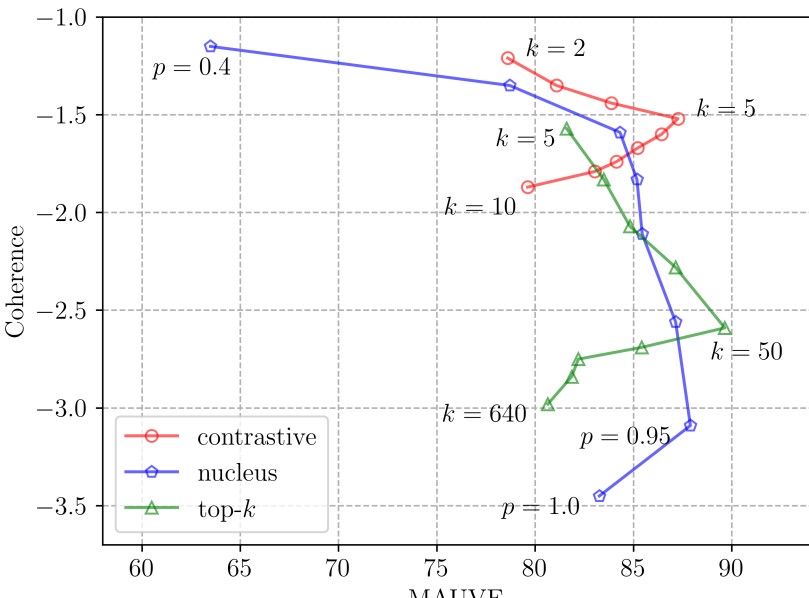

Figure 4: Contrastive search versus other sampling methods: (i) top-$k$; and (ii) nucleus sampling.

## 9 Conclusion and Future Work

In this work, we first investigate the isotropy of autoregressive LMs. Through extensive evaluations on LMs from 16 languages, we surprisingly find that the anisotropy problem *only* exists in the two specific English GPT-2-small/medium models. On the contrary, the rest of evaluated LMs are isotropic which is in contrast to the conclusion drawn by previous studies. Furthermore, based on our findings, we comprehensively evaluate contrastive search using *off-the-shelf* LMs on four generation tasks across 16 languages. Extensive human and automatic evaluations verify that contrastive search outperforms existing decoding methods by significant margins. More notably, on 12 out of the 16 evaluated languages, contrastive search performs comparably with human-level performances as judged by human evaluations.

For future work, we would like to suggest two research directions based on our study.

- **Open-domain knowledge probing of LMs**: Previous approaches (Petroni et al., 2019; Meng et al., 2022) for probing knowledge from LMs mainly focus on a fixed set of knowledge ontologies. Differently, contrastive search opens up another viable direction in which the world knowledge of the LMs with respect to a specific entity can be elicited through open-domain generation. In Appendix A.1, we provide an example on how to directly generate the factual knowledge of *"DeepMind Company"* from the LMs using contrastive search.

- **Dataset synthesization**: There has been a rising trend in using generative LMs to synthesize training data, therefore alleviating issues like data sparsity. By default, previous studies (Schick & Schütze, 2021; Ye et al., 2022) use sampling methods to create synthetic data. However, it still remains as an open question on how the choice of decoding method affects the system's downstream performances. We hypothesize that replacing sampling methods with contrastive search could further improve the quality of synthetic data, therefore benefiting the performances of downstream systems.

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

# Appendix

## Table of Contents

# A   More Generation Examples

## A.1   Example One

| Method | Generated Result |
|---|---|
| nucleus | **DeepMind Company is** a Cardiff-based start-up with an exclusive mission to build the world's largest ever deep-learning system to analyse the world's digital content and in particular, super-sized image content.

The system, the largest in the world with no previous expertise in image or digital content detection, will have previously relied on a mixture of machine learning, artificial neural networks, and storage, processing and retrieval techniques.

The AI system, called ImageNet, will take new approach to our challenge of data science and machine learning, significantly improving efficiency, natural language processing and full understanding of complex, high-dimensional images, with an Eye of the Tiger framework for extracting techniques to ensure correct detection of particular images in complex scenes.

Dr. Mark Ward, Dr. Alex Kudle, Dr. Ralph Pinchbeck and CTO, DeepMind Dr. Alex Kudle

Case Study: Derpy's Most Wanted: Fighting Cybersecurity, building a robot-aided smuggling network

InfoSec News, 06/07/2017

Dimitrios Papadimitriou (left) and Chris Bardy (right) at G+ XE, July 2017

How to model an industrial malware botnet

In this case study, we show how to build a deep-learning environment to model a new, massive ransomware botnet. Our model computes the distribution of user credentials stored on infected machines and produces a toolkit for open-source "modeling-as-code" (MATC) simulation. We elaborate on the resource management aspect of the toolkit, and how it can be adapted to working offline on embedded or cloud-based networks... |
| contrastive | **DeepMind Company is** a leader in artificial intelligence (AI). We have a long history of working with companies such as Google, Facebook, Amazon, and Microsoft to build products that improve people's lives, and today we are excited to announce that DeepMind's AlphaGo program has won the game of Go, becoming the first program to defeat a professional Go player.

The victory is a testament to the power of deep learning, and to the incredible work of our research team, which has been at the forefront of AI research for the past five years. AlphaGo is one of the most advanced Go programs ever created, and its performance is an important step towards the goal of human-level AI.

"This is the culmination of a decade of hard work," said Andy Ng, co-founder and CTO of DeepMind. "We are thrilled to have achieved this milestone and look forward to continuing to develop AI that can be used in a wide range of applications and to help people live better lives."

DeepMind's work on Go began in 2010, when it began to train a neural network to play Go using millions of games played by top Go players around the world. Since then, the team has refined the algorithm, adding more and more layers of reinforcement learning to make it better at recognizing patterns and making decisions based on those patterns. In the past year and a half, the team has made significant progress in the game, winning a record-tying 13 games in a row to move into the top four of the world rankings.

"The game of Go is a complex game in which players have to be very careful not to overextend their territory, and this is something that we have been able to improve over and over again," said Dr. Demis Hassabis, co-founder and Chief Scientific Officer of DeepMind. "We are very proud of our team's work, and we hope that it will inspire others to take the next step in their research and apply the same techniques to other problems."... |

Table 8: The documents generated by GPT-2-large using different methods given the prefix *"DeepMind Company is"*. The content that is irrelevant to the prefix is highlight in red. The factually correct and wrong content are highlighted in blue and violet, respectively. (best viewed in color)

Table 8 compares the generated results from GPT-2-large using different decoding methods given the prefix *"DeepMind Company is"*. On the one hand, the text generated by nucleus sampling contains factually incorrect content, and it quickly goes off-the-topic and produces text that is irrelevant to the prefix. On the other hand, the whole text generated by contrastive search well maintains the factual correctness as well as the semantic coherence.

### A.2 Example Two

---

**In a shocking finding, scientist discovered a herd of unicorns living in a remote, previously unexplored valley, in the Andes Mountains. Even more surprising to the researchers was the fact that the unicorns spoke perfect English.**

According to the BBC, a team of scientists led by Dr David MacKay, from the University of Bristol, spent two years searching for the unicorn herd, which they discovered during a survey of the area.

"It's a very rare find," MacKay told the BBC. "There are a few in the Himalayas, but this is the first time we've been able to find one in such a remote area."

The team was surprised to find a herd of unicorns living in a region that has been known to be a hotbed of poaching, with many of the animals poached for their horns, which are used in traditional Chinese medicine to treat everything from rheumatism to cancer.

"We knew that the area was rich in rhino horn, but we had no idea how many there were, or what they were doing there," MacKay said. "This is an area of high poaching pressure, and we wanted to find out what was going on."

In order to do so, the team used GPS collars to track the animals as they moved around the mountain and the surrounding area. The GPS data was then compared with information gathered from local villagers, who had a wealth of information about the animals' movements, including where they were eating, what they were doing at night, and how much time they spent in the mountains each day.

After analyzing the data, the team determined that the herd consisted of at least three species of unicorns, including a male and two females. One of the females was the mother of the male, and the other two were her daughters. All three had the same horn color, which is believed to be a sign of purity in the animal kingdom.

While the discovery is exciting, it's not the first time scientists have discovered an animal that speaks English. Last year, scientists discovered a species of porcupine that can be heard by humans, and has been dubbed "Porcupine Man" for his ability to converse with the human race.

---

Table 9: Another example generated by GPT-2-large using contrastive search.

Table 9 presents another example, with the length over hundreds of tokens, generated by GPT-2-large using contrastive search. Specifically, we use the prompt from the original OpenAI blog[17] which open-sourced GPT-2. Again, we see that contrastive search is able to generate a long document with coherent semantics and structure, revealing its clear advantages over existing decoding methods.

## B    Visualization on the Isotropy of GPT-2 Models.

To better understand the isotropy of GPT-2 models, we visually compare the representation space of different *off-the-shelf* GPT-2 models. Specifically, for each model, we compute the token representations using the same sentence *"Cambridge is a beautiful city.".* Then, we visualize the cosine similarity matrix of the output token representations as in Figure 5. On the one hand, we see that the representation space of the small (i.e. Figure 5(a)) and medium (i.e. Figure 5(b)) GPT-2 models are severely anisotropic, and their token representations reside in a narrow cone of the entire space with token cosine similarities over 0.90. On the other hand, the representation space of large (i.e. Figure 5(c)) and xl (i.e. Figure 5(d)) GPT-2 models are evenly distributed and clearly isotropic, which is also demonstrated by our results in Figure 1.

---

[17]https://openai.com/blog/better-language-models/

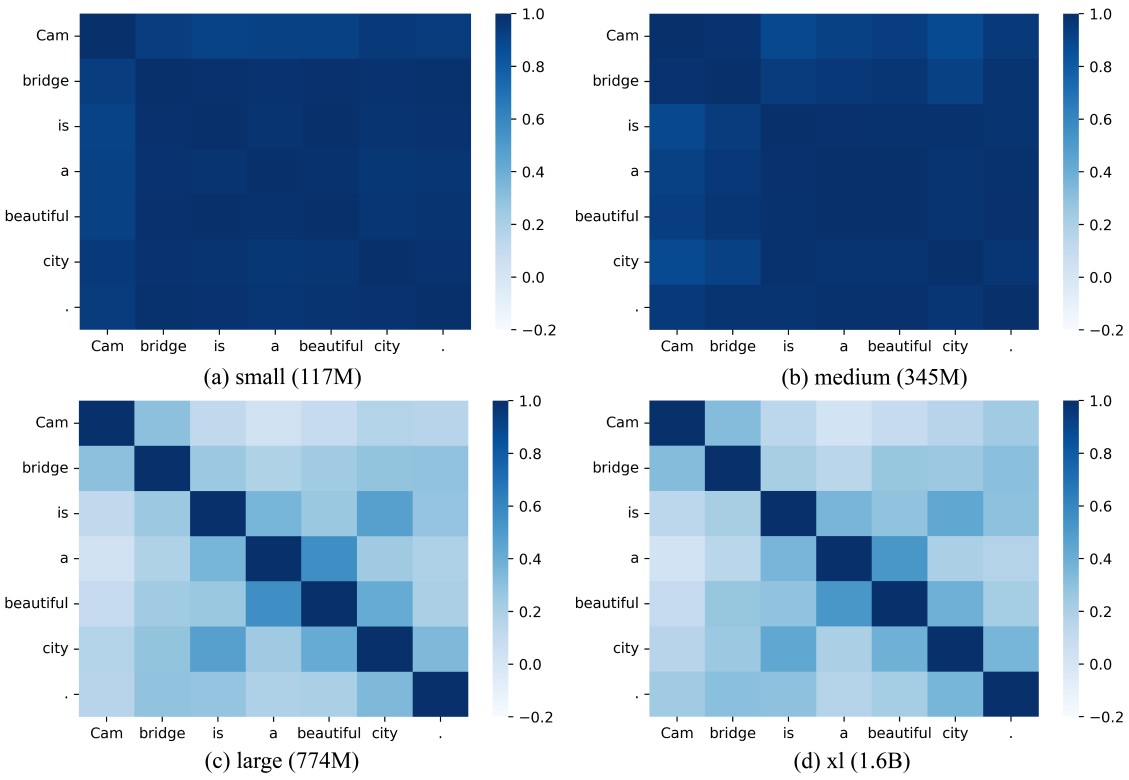

Figure 5: Visualizations on the token similarity matrix of different GPT-2 models. The token representations of different GPT-2 models are computed using the sentence, i.e., *"Cambridge is a beautiful city."*.

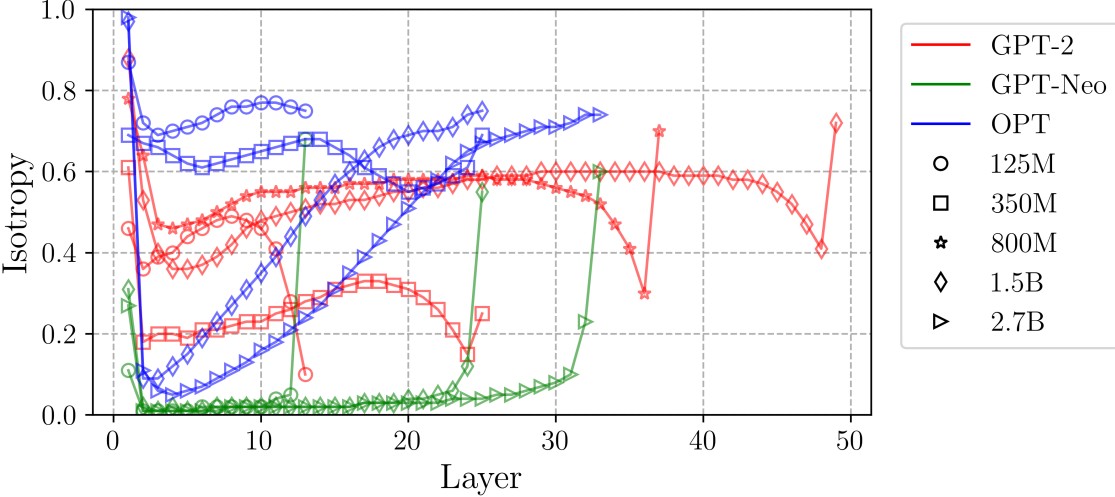

Figure 6: Layer-wise isotropy of different LMs.

## C   Layer-wise Isotropy of LMs

In this part, we investigate the isotropy of LMs in the intermediate layers. Specially, we use the token representations from the intermediate layers to compute the LM's isotropy following Eq. (2). Figure 6 plots the results of GPT-2, GPT-Neo, and OPT models with different scales. We see that the isotropy scores of intermediate layers in GPT-Neo models are generally lower than GPT-2's. On the other hand, for OPT models, the smaller models (e.g. OPT-125M) have higher isotropy than larger models (e.g. OPT-2.7B) in the intermediate layers. These different behaviours of different LMs echo with our remark in §3.1 that the

isotropy/anisotropy of LMs could relate to various factors (e.g. training data, number of model parameters, model initialization, optimization, and etc.). We leave the rigorous investigation on the unusual behaviours (i.e. anisotropy) of the GPT-2-small/medium models to our future work.

## D  Complete List of Evaluated LMs

In Table 10, we list our evaluated LMs across 16 languages, including the model scale and the corresponding isotropy score as presented in Section §3.1 and §3.2, respectively. To ensure the reproducibility of our results, all our evaluated LMs are publicly available in the Huggingface library (Wolf et al., 2019).

| Language | Code | HuggingFace Model Card | Size | Isotropy |
|---|---|---|---|---|
| English | en | https://huggingface.co/gpt2 | 117M | 0.10 |
| | | https://huggingface.co/gpt2-medium | 345M | 0.25 |
| | | https://huggingface.co/gpt2-large | 774M | 0.70 |
| | | https://huggingface.co/gpt2-xl | 1.6B | 0.72 |
| | | https://huggingface.co/EleutherAI/gpt-neo-125M | 125M | 0.68 |
| | | https://huggingface.co/EleutherAI/gpt-neo-1.3B | 1.3B | 0.55 |
| | | https://huggingface.co/EleutherAI/gpt-neo-2.7B | 2.7B | 0.60 |
| | | https://huggingface.co/facebook/opt-125m | 125M | 0.75 |
| | | https://huggingface.co/facebook/opt-350m | 350M | 0.69 |
| | | https://huggingface.co/facebook/opt-1.3b | 1.3B | 0.75 |
| | | https://huggingface.co/facebook/opt-2.7b | 2.7B | 0.74 |
| | | https://huggingface.co/facebook/opt-6.7b | 6.7B | 0.70 |
| | | https://huggingface.co/facebook/opt-13b | 13B | 0.66 |
| | | https://huggingface.co/facebook/opt-30b | 30B | 0.68 |
| Spanish | es | https://huggingface.co/datificate/gpt2-small-spanish | 117M | 0.77 |
| | | https://huggingface.co/DeepESP/gpt2-spanish-medium | 345M | 0.76 |
| French | fr | https://huggingface.co/asi/gpt-fr-cased-small | 117M | 0.76 |
| Portuguese | pt | https://huggingface.co/pierreguillou/gpt2-small-portuguese | 117M | 0.77 |
| Thai | th | https://huggingface.co/flax-community/gpt2-base-thai | 117M | 0.74 |
| Japanese | ja | https://huggingface.co/colorfulscoop/gpt2-small-ja | 117M | 0.72 |
| Korean | ko | https://huggingface.co/skt/kogpt2-base-v2/tree/main | 117M | 0.58 |
| | | https://huggingface.co/skt/ko-gpt-trinity-1.2B-v0.5 | 1.6B | 0.68 |
| Chinese | zh | https://huggingface.co/uer/gpt2-chinese-cluecorpussmall | 117M | 0.66 |
| Indonesian | id | https://huggingface.co/cahya/gpt2-small-indonesian-522M | 117M | 0.66 |
| | | https://huggingface.co/flax-community/gpt2-medium-indonesian | 345M | 0.67 |
| | | https://huggingface.co/cahya/gpt2-large-indonesian-522M/ | 774M | 0.81 |
| Bengali | bn | https://huggingface.co/flax-community/gpt2-bengali | 117M | 0.62 |
| Hindi | hi | https://huggingface.co/surajp/gpt2-hindi | 117M | 0.62 |
| Arabic | ar | https://huggingface.co/akhooli/gpt2-small-arabic | 117M | 0.53 |
| | | https://huggingface.co/aubmindlab/aragpt2-medium | 345M | 0.64 |
| German | de | https://huggingface.co/ml6team/gpt2-small-german-finetune-oscar | 117M | 0.83 |
| | | https://huggingface.co/ml6team/gpt2-medium-german-finetune-oscar | 345M | 0.81 |
| Dutch | nl | https://huggingface.co/ml6team/gpt2-small-dutch-finetune-oscar | 117M | 0.80 |
| | | https://huggingface.co/ml6team/gpt2-medium-dutch-finetune-oscar | 345M | 0.79 |
| Russian | ru | https://huggingface.co/sberbank-ai/rugpt3small_based_on_gpt2 | 117M | 0.67 |
| | | https://huggingface.co/sberbank-ai/rugpt3medium_based_on_gpt2 | 345M | 0.72 |
| | | https://huggingface.co/sberbank-ai/rugpt3large_based_on_gpt2 | 774M | 0.77 |
| Italian | it | https://huggingface.co/LorenzoDeMattei/GePpeTto | 117M | 0.69 |

Table 10: The complete list of the evaluated languages as well as the corresponding LMs.

# E  Evaluation Setups of Multilingual Open-ended Text Generation

Table 11 presents the details of (i) our evaluated languages; (ii) the link address of assessed LMs; and (iii) the hyperparameters (i.e. $k$ and $\alpha$) used in contrastive search for our experiments in multilingual open-ended text generation.

| Language | HuggingFace Model Card | $k$ | $\alpha$ |
|---|---|---|---|
| English | https://huggingface.co/gpt2-large | 4 | 0.6 |
| Russian | https://huggingface.co/sberbank-ai/rugpt3large_based_on_gpt2 | 4 | 0.6 |
| Indonesian | https://huggingface.co/cahya/gpt2-small-indonesian-522M | 3 | 0.6 |
| Spanish | https://huggingface.co/datificate/gpt2-small-spanish | 3 | 0.6 |
| German | https://huggingface.co/ml6team/gpt2-medium-german-finetune-oscar | 4 | 0.6 |
| Dutch | https://huggingface.co/ml6team/gpt2-medium-dutch-finetune-oscar | 4 | 0.6 |
| Korean | https://huggingface.co/skt/ko-gpt-trinity-1.2B-v0.5 | 3 | 0.6 |
| Arabic | https://huggingface.co/akhooli/gpt2-small-arabic | 3 | 0.6 |
| French | https://huggingface.co/asi/gpt-fr-cased-small | 3 | 0.6 |
| Portuguese | https://huggingface.co/pierreguillou/gpt2-small-portuguese | 3 | 0.6 |
| Thai | https://huggingface.co/flax-community/gpt2-base-thai | 3 | 0.6 |
| Japanese | https://huggingface.co/colorfulscoop/gpt2-small-ja | 5 | 0.6 |
| Chinese | https://huggingface.co/uer/gpt2-chinese-cluecorpussmall | 3 | 0.6 |
| Bengali | https://huggingface.co/flax-community/gpt2-bengali | 3 | 0.6 |
| Hindi | https://huggingface.co/surajp/gpt2-hindi | 3 | 0.6 |
| Italian | https://huggingface.co/LorenzoDeMattei/GePpeTto | 3 | 0.6 |

Table 11: The language models that we use in the experiments of multilingual open-ended text generation. The hyperparameters (i.e. $k$ and $\alpha$) of contrastive search used for different LMs are also provided.

## F  Detailed Results on Document Summarization

Table 12 presents the detailed evaluation results on XSum dataset.

| Shot | Method | Run | OPT-125M | | | OPT-350M | | | OPT-1.3B | | | OPT-2.7B | | |
|---|---|---|---|---|---|---|---|---|---|---|---|---|---|---|
| | | | R-1 | R-2 | R-L | R-1 | R-2 | R-L | R-1 | R-2 | R-L | R-1 | R-2 | R-L |
| One | beam | 1 | 13.71 | 1.01 | 10.17 | 17.82 | 2.69 | 13.44 | 23.19 | 5.69 | 18.11 | 22.34 | 5.41 | 17.33 |
| | | 2 | 14.37 | 1.70 | 10.82 | 15.23 | 2.04 | 11.52 | 23.77 | 6.05 | 18.35 | 26.28 | 7.42 | 20.45 |
| | | 3 | 12.37 | 1.12 | 9.53 | 13.46 | 1.40 | 9.86 | 23.15 | 5.70 | 17.75 | 26.35 | 7.92 | 20.71 |
| | | ave. | 13.48 | 1.28 | 10.17 | 15.50 | 2.04 | 11.61 | 23.37 | 5.81 | 18.07 | 24.99 | 6.92 | 19.50 |
| | | (std.) | (0.83) | (0.30) | (0.53) | (1.79) | (0.53) | 1.46 | (0.28) | (0.17) | (0.25) | (1.87) | (1.08) | (1.54) |
| | nucleus | 1 | 12.19 | 0.81 | 9.30 | 12.72 | 0.76 | 9.80 | 16.10 | 1.85 | 12.09 | 19.39 | 3.37 | 14.82 |
| | | 2 | 12.35 | 0.84 | 9.54 | 12.45 | 0.76 | 9.46 | 14.24 | 1.42 | 10.75 | 15.11 | 1.85 | 11.65 |
| | | 3 | 12.55 | 0.88 | 9.63 | 11.65 | 0.78 | 8.89 | 16.15 | 2.12 | 11.99 | 19.93 | 3.88 | 15.00 |
| | | ave. | 12.36 | 0.84 | 9.49 | 12.27 | 0.77 | 9.38 | 10.01 | 1.80 | 11.61 | 18.14 | 3.03 | 13.82 |
| | | (std.) | (0.15) | (0.03) | (0.14) | (0.45) | (0.01) | (0.38) | (6.00) | (0.29) | (0.61) | (2.16) | (0.86) | (1.54) |
| | contrastive | 1 | 18.17 | 2.53 | 13.57 | 21.30 | 4.04 | 16.30 | 26.83 | 7.22 | 21.04 | 27.21 | 7.91 | 21.60 |
| | | 2 | 17.17 | 2.07 | 13.06 | 16.90 | 2.23 | 12.80 | 24.88 | 6.01 | 18.74 | 26.27 | 7.24 | 20.21 |
| | | 3 | 12.23 | 1.27 | 9.46 | 13.70 | 1.73 | 10.70 | 25.55 | 6.49 | 19.49 | 29.84 | 9.51 | 23.50 |
| | | ave. | **15.86** | **1.96** | **12.03** | **17.30** | **2.67** | **13.27** | **25.75** | **6.57** | **19.76** | **27.77** | **8.22** | **21.77** |
| | | (std.) | (2.60) | (0.52) | (1.83) | (3.12) | (0.99) | (2.31) | (0.81) | (0.50) | (0.96) | (1.51) | (0.95) | (1.35) |
| Two | beam | 1 | 16.68 | 2.15 | 12.84 | 16.96 | 2.29 | 13.26 | 27.10 | 8.20 | 21.54 | 26.88 | 8.28 | 21.45 |
| | | 2 | 17.44 | 2.43 | 13.20 | 18.62 | 3.15 | 14.25 | 24.18 | 6.40 | 18.70 | 24.45 | 7.04 | 19.29 |
| | | 3 | 16.95 | 1.47 | 12.88 | 17.39 | 1.45 | 13.24 | 24.81 | 6.69 | 18.92 | 26.22 | 7.84 | 20.51 |
| | | ave. | 17.02 | 2.02 | 12.97 | 17.66 | 2.30 | 13.58 | 25.36 | 7.10 | 19.72 | 25.85 | 7.72 | 20.42 |
| | | (std.) | (0.31) | (0.40) | (0.16) | (0.70) | (0.69) | (0.47) | (1.25) | (0.79) | (1.29) | (1.03) | (0.51) | (0.88) |
| | nucleus | 1 | 13.01 | 0.93 | 9.80 | 12.32 | 0.97 | 9.62 | 20.16 | 3.60 | 15.21 | 22.25 | 4.95 | 16.99 |
| | | 2 | 12.27 | 0.82 | 9.47 | 11.78 | 0.83 | 9.23 | 14.07 | 1.40 | 10.97 | 13.60 | 1.29 | 10.69 |
| | | 3 | 12.97 | 0.86 | 9.86 | 13.25 | 1.16 | 10.22 | 19.74 | 3.66 | 14.89 | 21.37 | 4.48 | 16.24 |
| | | ave. | 12.75 | 0.87 | 9.71 | 12.45 | 0.99 | 9.69 | 17.99 | 2.89 | 13.69 | 19.07 | 3.57 | 14.64 |
| | | (std.) | (0.34) | (0.05) | (0.17) | (0.61) | (0.14) | (0.41) | (2.78) | (1.05) | (1.93) | (3.89) | (1.63) | (2.81) |
| | contrastive | 1 | 17.17 | 2.48 | 13.33 | 18.97 | 3.32 | 14.72 | 29.10 | 8.61 | 23.00 | 31.10 | 10.51 | 24.92 |
| | | 2 | 18.98 | 3.17 | 14.78 | 18.99 | 3.28 | 14.56 | 24.14 | 5.80 | 18.68 | 24.40 | 6.38 | 19.33 |
| | | 3 | 17.97 | 2.24 | 13.56 | 18.57 | 2.43 | 14.15 | 28.70 | 8.20 | 21.80 | 31.56 | 10.38 | 24.95 |
| | | ave. | **18.04** | **2.63** | **13.89** | **18.84** | **3.01** | **14.48** | **27.31** | **7.54** | **21.16** | **29.02** | **9.09** | **23.07** |
| | | (std.) | (0.74) | (0.39) | (0.64) | (0.19) | (0.41) | (0.24) | (2.25) | (1.24) | (1.82) | (3.27) | (1.92) | (2.64) |

Table 12: Detailed results on XSum benchmark over different selections of in-context examples.

## G  Detailed Results on Code Generation

Table 13 presents the detailed results of nucleus sampling on HumanEval dataset.

| Model | run-1 | run-2 | run-3 | average | (std.) |
|---|---|---|---|---|---|
| CodeGen-350M-mono | 4.88 | 6.10 | 4.27 | 5.08 | (0.76) |
| CodeGen-2B-mono | 10.98 | 11.59 | 10.37 | 10.98 | (0.50) |

Table 13: Detailed pass rate@1 (%) results of nucleus sampling on code generation.

## H  Machine Translation

Table 14 presents the detailed results on IWSLT14 De-En dataset.

| Shot | Method | run | OPT-125M | OPT-350M | OPT-1.3B | OPT-2.7B |
|------|--------|-----|----------|----------|----------|----------|
| One | beam | 1 | 0.00 | 0.08 | 4.09 | 13.16 |
| | | 2 | 0.00 | 0.00 | 7.30 | 14.76 |
| | | 3 | 0.00 | 0.00 | 5.19 | 14.27 |
| | | ave. | 0.00 | **0.03** | 5.53 | **14.06** |
| | | (std.) | (0.00) | (0.04) | (1.33) | (0.67) |
| | nucleus | 1 | 0.00 | 0.00 | 1.51 | 6.29 |
| | | 2 | 0.00 | 0.00 | 3.11 | 7.84 |
| | | 3 | 0.00 | 0.00 | 1.91 | 7.32 |
| | | ave. | 0.00 | 0.00 | 2.18 | 7.15 |
| | | (std.) | (0.00) | (0.00) | (0.68) | (0.64) |
| | contrastive | 1 | 0.00 | 0.00 | 5.61 | 11.93 |
| | | 2 | 0.00 | 0.00 | 8.84 | 13.75 |
| | | 3 | 0.14 | 0.00 | 6.86 | 13.26 |
| | | ave. | **0.05** | 0.00 | **7.10** | 12.98 |
| | | (std.) | (0.07) | (0.00) | (1.33) | (0.77) |
| Few | beam | 1 | 0.00 | 0.23 | 7.58 | 14.10 |
| | | 2 | 0.00 | 0.00 | 9.41 | 14.59 |
| | | 3 | 0.00 | 0.00 | 8.64 | 15.07 |
| | | ave. | 0.00 | **0.08** | **8.54** | **14.59** |
| | | (std.) | (0.00) | (0.11) | (0.75) | (0.40) |
| | nucleus | 1 | 0.10 | 0.00 | 3.30 | 7.94 |
| | | 2 | 0.00 | 0.10 | 4.90 | 8.60 |
| | | 3 | 0.00 | 0.00 | 4.47 | 8.54 |
| | | ave. | 0.03 | 0.03 | 4.22 | 8.36 |
| | | (std.) | (0.05) | (0.05) | (0.68) | (0.30) |
| | contrastive | 1 | 0.15 | 0.15 | 7.39 | 13.36 |
| | | 2 | 0.00 | 0.00 | 8.93 | 13.50 |
| | | 3 | 0.00 | 0.00 | 8.86 | 13.70 |
| | | ave. | **0.05** | 0.05 | 8.39 | 13.52 |
| | | (std.) | (0.07) | (0.07) | (0.71) | (0.14) |

Table 14: Detailed evaluation results on IWSLT14 De-En dataset.

## I  Machine Translation with Encoder-Decoder Models

In this section, we extend our evaluation on machine translation task to encoder-decoder models. Same as in §7, we use IWSLT14 dataset as our evaluation benchmark and we consider the translation task from both directions, i.e. De-to-En and En-to-De. For the encoder-decoder models, we use the publicly available translation models (Tiedemann & Thottingal, 2020) in both De-to-En[18] and En-to-De[19] directions.

Following §7, we generate translations using different decoding methods, including beam search, nucleus sampling, and contrastive search. The generated results are evaluated from two aspects: (i) BLEU; and (ii) BERTScore (F1) (Zhang et al., 2019). In addition, we also report the decoder-side isotropy (see Eq (2)) of the encoder-decoder models using texts from the target language.

---

[18]https://huggingface.co/Helsinki-NLP/opus-mt-de-en
[19]https://huggingface.co/Helsinki-NLP/opus-mt-en-de

| Method | De-to-En | | | En-to-De | | |
|---|---|---|---|---|---|---|
| | BLEU | BERTScore | isotropy | BLEU | BERTScore | isotropy |
| beam | **33.98** | **0.95** | | 28.36 | **0.86** | |
| nucleus | 30.22 $(\pm 0.53)$ | 0.93 $(\pm 0.01)$ | 0.53 | 26.99 $(\pm 0.33)$ | 0.84 $(\pm 0.01)$ | 0.55 |
| contrastive | 32.61 | **0.95** | | **28.49** | **0.86** | |

Table 15: Evaluation results on IWSLT14 dataset.

Table 15 presents the experimental results. First, we see that the BLEU and BERTScore of contrastive search are comparable to the ones obtained by beam search. Surprisingly, contrastive search even obtains a slightly better BLEU score than beam search on the En-to-De translation task. Second, the decoder-side isotropy scores suggest that the encoder-decoder models display a high level of isotropy same as the autoregressive models (§3), making contrastive search directly applicable. We leave the isotropy analysis of other models from the encoder-decoder family to our future work.

## J   Ablation Study on the Hyperparameters of Contrastive Search

In this section, we present a detailed ablation study on the hyperparameters (i.e., $k$ and $\alpha$ in Eq. (3)) of contrastive search. We follow §8.2 and generate text using GPT-2-large with contrastive search. Specifically, we simultaneously vary the value of $k$ and $\alpha$, i.e. $k$ is chosen from $\{2, 5, 10\}$ and $\alpha$ is chosen from 0.1 to 0.9.

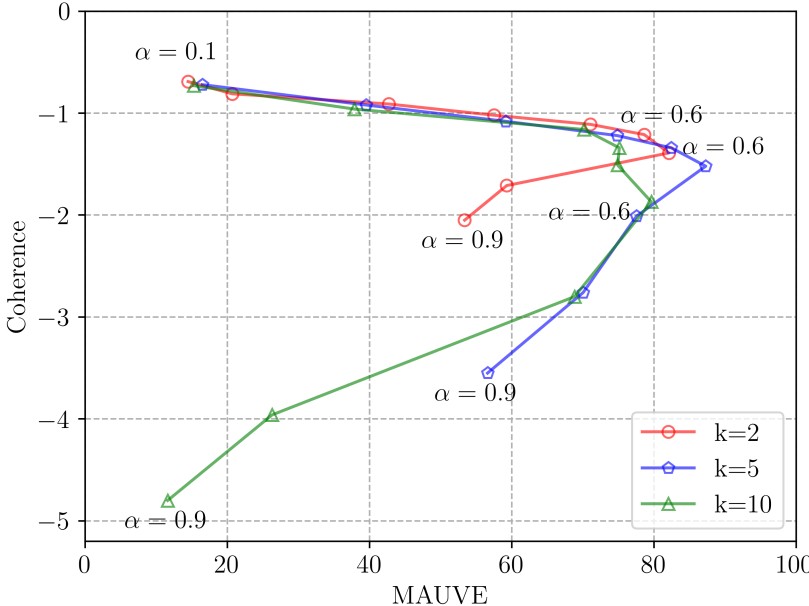

Figure 7: Ablation study on the hyperparameters of contrastive search.

Same as in §8.2, the generated texts are evaluated from two aspects: MAUVE and coherence (obtained with the OPT-2.7B model). Figure 7 plots the results of different hyperparameters. We see that, when $k$ is constant, increasing $\alpha$ generally decreases the coherence score of the generated text. In contrast, the MAUVE score of the generated text increases when changing $\alpha$ from 0.1 to 0.6. On the other hand, when $\alpha$ is from 0.6 to 0.9, the MAUVE score decreases. In general, for different $k$, the overall trends are relatively the same and the value of $\alpha$ has larger impact on the generated results.

## K    Correlation Study between Isotropy and Variance of Degeneration Penalty

In this section, we further investigate the correlation between the LM's isotropy and the variance of degeneration penalty. We conduct experiments using different LMs (i.e. GPT-2, GPT-Neo, and OPT) with various scales (up to 2.7b). Specifically, following §8.1, we use the LM to generate text (up to 200 tokens) conditioned on the prefix texts (restricted to 40 tokens) from the held-out set of WebText. For all LMs, the $k$ and $\alpha$ in contrastive search are set as 5 and 0.6, respectively.

As for evaluation, we measure the variance of degeneration penalty averaged over the entire decoding process and the measurement $s$ is defined as

$$s = \frac{1}{T}\sum_{t=1}^{T} f(t; \theta, \mathcal{D}),\tag{7}$$

where $T$ is the generation length (i.e. 200 in our experiments) and $f(t; \theta, \mathcal{D})$ is defined in Eq. (6).

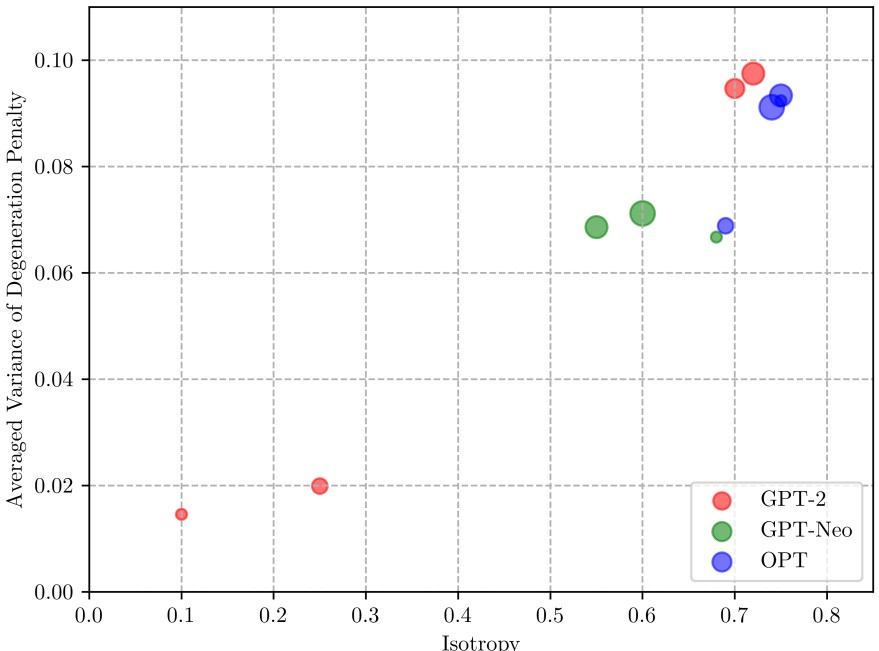

Figure 8: Correlation between the LM's isotropy and the averaged variance of degeneration penalty. The circle size corresponds to the scale of the LM.

Figure 8 plots the experimental results from which we can clearly observe a positive correlation between the LM's isotropy and the averaged variance of degeneration penalty. These results further demonstrate that a high isotropy of the LM is desirable as it improves the variance of degeneration penalty, therefore benefiting the performance of contrastive search.

## L    Comparison between Off-the-shelf and SimCTG using Contrastive Search

In previous sections, we have demonstrated that contrastive search works well on off-the-shelf LMs. In this part, our goal is to compare the performance of contrastive search using off-the-shelf LM and LM trained with SimCTG (Su et al., 2022b). To this end, we follow Su et al. (2022b) and conduct experiments on the Wikitext-103 benchmark (Merity et al., 2017). To make a fair comparison between both models (i.e. Off-the-shelf and SimCTG), we fine-tune the GPT-2-large model on Wikitext-103 for the same training steps (i.e. 40k). Specifically, the off-the-shelf LM is fine-tuned with the MLE objective which is originally used to pre-train the LM as

$$\mathcal{L}_{\text{MLE}} = -\frac{1}{|\boldsymbol{x}|}\sum_{i=1}^{|\boldsymbol{x}|}\log p_{\theta}(x_i|\boldsymbol{x}_{<i}),\tag{8}$$

where $\theta$ is the LM and $\boldsymbol{x}$ is a variable-length text sequence. The other compared model, i.e. SimCTG, is obtained by fine-tuning the LM with the SimCTG objective which is proposed by Su et al. (2022b). Following Su et al. (2022b), the length of the prefix text is set as 32 and the maximum generated length is set as 128. For both models, the $k$ and $\alpha$ in contrastive search are set as 5 and 0.6, respectively.

| Model | diversity(%)↑ | MAUVE(%)↑ | gen-length | coherence↑ |
|---|---|---|---|---|
| SimCTG | 92.24 | 85.32 | 110.34 | **-1.43** |
| Off-the-shelf | **92.48** | **85.66** | 108.97 | -1.46 |

Table 16: Automatic evaluation results on Wikitext-103. ↑ means the higher the better.

| Method A is Better | | Neutral | Method B is Better | |
|---|---|---|---|---|
| Off-the-shelf | **28.2**%$^{\parallel}$ | 44.9% | 26.9%$^{\parallel}$ | SimCTG |

Table 17: Human evaluation results on Wikitext-103. $^{\parallel}$ means one method performs comparably with the other with $p$-value $> 0.4$.

**Evaluation Results.** We follow §4.1.1 and §4.1.2, and evaluate the two compared models through automatic and human evaluations. The evaluated results are presented in Table 16 and 17, respectively. We can see that Off-the-shelf LM performs comparably with SimCTG on both evaluations. These results suggest that, when the LM (e.g. GPT-2-large) is intrinsically isotropic, the additional training of SimCTG may not be necessary for contrastive search to work well. On the other hand, when the LM is anisotropic (e.g. GPT-2-small), the training of SimCTG is indispensable as demonstrated by previous work (Su et al., 2022b).

