# OpenReview forum: "Contrastive Search Is What You Need For Neural Text Generation"
_TMLR — Accepted by TMLR_

### Review · Reviewer_XsZM · 2022-11-25

**Summary Of Contributions:**

This work has two contributions, one regarding the isotropic nature of pre-trained language models and the other comparing contrastive search with other inference methods. For the analysis of isotropy, this work introduce a document averaged metric based on the average token cosine similarity within a sentence. One of the findings is that the most pre-trained language models are isotropic, but only smallish GPT-2 pre-trained English models are anisotropic. In the systematic comparisons of various inference methods, this paper shows that the previous proposed contrastive search is at least as comparable to other methods, e.g., typical, top-k and nucleus sampling, under objective metrics in open-ended text generation tasks. Subjective evaluation through pair-wise comparison shows clear gains. The inference methods are also compared in other tasks, i.e., summarization, code generation and MT, and this work shows that the contrastive search performs better than others.

**Audience:**

Yes

**Broader Impact Concerns:**

I have no concerns to this submission.

**Claims And Evidence:**

Yes

**Requested Changes:**

Major comment:

- It is not clear why smallish English GPT-2 models suffer from an anisotropic issue.

  Analysis on other layers might give us a clue given that there were similar analysis on BERT, e.g., https://arxiv.org/abs/2001.08950 and https://openreview.net/forum?id=dUV91uaXm3, showing that token representations are similar at the higher layers.

- BLEU scores in machine translation experiments are far below SOTA and it might not be meaningful to make comparisons.

  Although the focus is on pre-trained language models, I feel it would be better to run experiments for encoder-decoder for a fair comparison together with isotropic/anisotropic analysis. It would be also good to report non-BLEU scores, e.g., BERTScore and/or COMET, since those scores might have different tendencies.

- I want to see if there's any tradeoff by varying $\alpha$ for contrastive search in Figure 4.

Others

- h_v in Equation 3 is the representation at time t which is used to predict the token at time t+1 if my understanding is correct. I feel better to make is explicity.

- Please explain how to interpret Table 3 and 4. I suspect the column for system A might implies "win for system A" but there exists no clear description.

- Please fix Figure 4 in that top-k is cited as top-n.




**Strengths And Weaknesses:**

Strengths

- The analysis of the isotropic nature of pre-trained language models is interesting, and the finding might have an impact to the research community.
- Multiple well-known inference methods, i.e., greedy/beam search, typical/top-k/nucleus sampling and contrastive search, are systematically compared on various tasks, e.g., open-ended generation, summarization, code generation and machine translation. The findings in the open-ended generation shows that contrastive search is comparable to sampling methods under objective evaluation metrics, and better in human evaluation.
- The analysis of the contrastive search in section 8 is convincing in that the contrastive search is effective especially when a pre-trained language model is isotropic.

Weaknesses

- The analysis of the isotropy is an interesting contribution, though, it is not clear why small GPT-2 English models are anisotropic. I think this question is very important given that the isotropy is a key to the success of the contrastive search as demonstrated in section 8.1.
- Some analysis is a bit hard to follow. For example, little is explained regarding the human judgement results shown in Table 3 and 4, and thus, it is hard to interpret those tables.

---

> ### Author Response · Authors · 2022-11-29
> **Response to the Reviewer XsZM**
>
> Thank you for your thoughtful reviews and valuable suggestions!
>
> ### 1. Isotropy in the intermediate layers:
> In the Appendix C (i.e. Figure 6) of our revised manuscript, we plot the isotropy results in the intermediate layers of GPT-2, GPT-Neo, and OPT models with different scales. We observe that different models behave quite differently. For instance, the intermediate isotropy of GPT-Neo is much lower than the one of GPT-2. Moreover, even for the same model, different model scales could lead to different behaviours. For example, the smaller OPT models (e.g. OPT-125M) have higher isotropy than larger models (e.g. OPT-2.7B) in the intermediate layers.
>
> These different behaviours of different LMs echo our remark in Section 3.1 that the isotropic/anisotropic property of the LM could relate to various factors (e.g. training data, model scale, initialization, optimization, and etc.). Therefore, we leave the rigorous investigation on the unusual behaviours (i.e. anisotropy) of the GPT-2-small/medium models to our future work.
>
>
> ### 2. Machine translation with encoder-decoder models:
> We supplement the results of encoder-decoder models on the machine translation task in the Appendix I of our revised manuscript. Specifically, we conduct experiments on the IWSLT14 benchmark in both De-to-En and En-to-De directions. For the encoder-decoder models, we use the publicly available translation models [1] in both [De-to-En](https://huggingface.co/Helsinki-NLP/opus-mt-de-en) and [En-to-De](https://huggingface.co/Helsinki-NLP/opus-mt-en-de) directions.
>
> Following Section 7, we generate translations using different methods, including beam search, nucleus sampling, and contrastive search. The generated results are evaluated from two aspects: (i) BLEU; and (ii) BERTScore (F1). In addition, we also report the decoder-side isotropy of the encoder-decoder models using texts from the target language.
>
>
> |Method|BLEU|BERTScore|isotropy|
> |:-------------:|:-------------:|:-----------------------:|:-----------------------:|
> |beam|**33.98**|**0.95**|0.53|
> |nucleus|30.22|0.93|0.53|
> |contrastive|32.61|**0.95**|0.53|
>
> **Table 1: Evaluation Results on IWSLT14 De-to-En.**
>
> |Method|BLEU|BERTScore|isotropy|
> |:-------------:|:-------------:|:-----------------------:|:-----------------------:|
> |beam|28.36|**0.86**|0.55|
> |nucleus|26.99|0.84|0.55|
> |contrastive|**28.49**|**0.86**|0.55|
>
> **Table 2: Evaluation Results on IWSLT14 En-to-De.**
>
> Table 1 and 2 present the experimental results. First, we see that the BLEU and BERTScore of contrastive search are comparable to the ones obtained by beam search. Surprisingly, contrastive search even obtains a slightly better BLEU score than beam search on the En-to-De translation task. Second, the decoder-side isotropy scores suggest that the encoder-decoder models display a high level of isotropy same as the autoregressive models, making contrastive search directly applicable.
>
>
> ### 3. Trade-off between k and alpha in contrastive search:
> In the Appendix J (i.e. Figure 7) of our revised manuscript, we plot the trade-off between k and alpha in contrastive search. Our results show that, when k is constant, the increase of alpha generally decreases the coherence score of the generated text. In contrast, the MAUVE score of the generated text increases when changing alpha from 0.1 to 0.6. On the other hand, when alpha is from 0.6 to 0.9, the MAUVE score decreases. In general, for different k, the overall trends are relatively the same and the value of alpha has a larger impact on the generated results.
>
> ### 4. Other suggested changes:
> Thank you for your suggestions on the additional changes. We have revised our manuscript corresponding to your suggestions as below:
> * (1) **Clarification on h_v:** We have added a more detailed description of h_v in Section 2.2 as highlighted in red.
> * (2) **Descriptions of tables in human evaluations:** We have changed the column headers in the tables of human evaluations. Specifically, the column headers are changed to :(i) Method A is Better; (ii) Neutral; and (iii) Method B is Better. We hope the tables are more readable now.
> * (3) **Notation in Figure 4:** We have corrected the notation for top-k sampling in Figure 4.
>
> ### Reference:
> > [1] Tiedemann and Thottingal, 2020. "OPUS-MT — Building open translation services for the World"

---

> > ### Comment · Reviewer_XsZM · 2022-12-06
> > **Thanks for your response**
> >
> > Thanks for the updates and they look quite reasonable to me.
> >
> > One more very minor question is related to the cosine similarity of the end of sentence representation. In autoregressive models, the representation for the end of sentence token is basically undefined since it will not suffer any losses during training. I'm just curious how that will impact the inference procedure especially when determining the search termination. Probably it might have an impact for lengths, and Table 2 might indicate that the contrastive search tends to generate a bit shorter texts.

---

> > > ### Author Response · Authors · 2022-12-06
> > > **Thank you for reading our response!**
> > >
> > > Thank you very much for reading our response!
> > >
> > > Actually, during the pre-training procedure of the language model, the end-of-sequence token was added to indicate the boundary between different pre-training documents. So it should receive losses during the pre-training stage. That being said the end-of-sequence token should be a valid token and its representation is properly optimized. From Table 2, we see that the generated length of contrastive search is longer than beam search and nucleus sampling and is comparable to other baseline methods. This might also suggest that the end-of-sequence token is a valid token and does not impact the performance of contrastive search.
> > >
> > > Thank you again for reading our response. Please let us know if you have further questions or comments. Many thanks!
> > >
> > > The Authors

---

> > > > ### Comment · Reviewer_XsZM · 2022-12-06
> > > > **SGTM**
> > > >
> > > > Thanks for the clarification. I totally missed that the token is abused as a separator.

---

> > > > > ### Author Response · Authors · 2022-12-06
> > > > > **Many thanks!**
> > > > >
> > > > > No worries! Thank you again for reading our response!
> > > > >
> > > > > Many thanks!
> > > > >
> > > > > The Authors

---

### Review · Reviewer_BDTk · 2022-11-25

**Summary Of Contributions:**

This paper presents two main results related to contrastive search for neural text generation.

The first result shows that the vast majority of publicly-available language models, varying both size and language, are isotropic, meaning that self-similarity between token representation is low. This contradicts previous results built on the assumption that LMs are anisotropic: it turns out that only a small set of special cases (GPT2-small and medium) fall into the anisotropic category, and those happen to be the ones that were tested previously.

The second result is to test the previously-proposed contrastive search algorithm with a number of off-the-shelf LMs on a number of tasks. With the exception of MT (where beam search continues to dominate), the results are overwhelmingly positive, both as measured by automatic metrics and human evaluation.

Finally, as a third mini-contribution, they propose a coherence metric, measured by the averaged log likelihood of some other (larger) evaluator LM. This metric is necessary to see the improvements of contrastive search using automatic metrics.


**Audience:**

Yes

**Broader Impact Concerns:**

I have no broader impact concerns. No broader impact statement is present.

**Claims And Evidence:**

Yes

**Requested Changes:**

Of course, I would like to see the two claims called out about softened, and I would love to see a comparison between off-the-shelf and SimCTG.

Another suggestion would be to broaden Section 8.1 into a complete correlation study between isotropy and variance of degeneration penalty. It feels like the authors have data for many LMs, but opted to only evaluate gpt2-*.


**Strengths And Weaknesses:**

Strengths:

The paper’s first contribution is surprising given previous work and helps avoid having the field continue to advance under false assumptions.

The second contribution, that contrastive search works off the shelf with many LMs, helps reinforce the first and will help expand adoption of this previously-proposed method.

The paper is clear and easy to follow.

The evaluation is exhaustive, is non-trivial, and looks trustworthy. It is likely to save others compute cycles.

The extensive use of human evaluation makes it difficult to deny the improvements of contrastive search.

Weaknesses:

The paper seems to be making two points: (1) most LMs are isotropic, and therefore (2) you can use contrastive search on the LMs without specialized training. The absolute improvements over reasonable baselines (nucleus sampling, beam search, etc) are nice, but I feel like to drive the point home, they should have shown that for these isotropic LMs, they should have compared off-the-shelf+contrastive search with SimCTG+contrastive search. Of course, the paper is very careful to not claim that SimCTG is not helpful, instead they show that off-the-shelf works fine, but ignoring the question of whether SimCTG is still useful feels a little lazy.

The paper makes some claims that are pretty strong; chief among them: “Autoregressive LMs are naturally isotropic.” Given that they’ve just disproved a similar claim in the other direction, and they have two counter-examples (GPT-2-small and medium), this feels a little premature. With no attempt made to determine why those 2 are anisotropic when every other LM is not (this is left to future work in a Remark), it feels very risky to make such a broad statement. If some slightly different hyper-parameter configuration becomes popular, they may find themselves facing down a legion of anisotropic LMs.

The other strong claim is that the text output by contrastive search is “indistinguishable with one written by a human” - I feel like the field could do with having fewer claims like this.

The writing has problems in just a few spots:

2.2: “The second term, degeneration penalty, measures how discriminative of the candidate v with respect to the previous context x<t “ - this doesn’t make much sense.

3.1: Typo in the size of GPT-2-large.

4.1.1 Points (i) and (ii) flow nicely in a single sentence. Point (iii) starts a whole new paragraph: this is jarring.

---

> ### Author Response · Authors · 2022-11-29
> **Response to the Reviewer BDTk**
>
> Thank you for your thoughtful reviews and constructive suggestions!
>
> ### 1. Two claims to be softened:
> We have softened both claims in our revised manuscript. Specifically, the revisions regarding "Autoregressive LMs are naturally isotropic" can be found in the (1) Introduction section; (2) Section 3.1.; (3) Section 3.2.; and (4) Conclusion section. The revision regarding "indistinguishable with one written by a human" can be found in Section 4.1.2. All revisions are highlighted in red.
>
> ### 2. Comparison between off-the-shelf and SimCTG.
> In the Appendix L of our revised manuscript, we compare the performance of contrastive search using off-the-shelf and SimCTG based on the GPT-2-large model. We follow the same experimental setup as in [1] and evaluate both models on the Wikitext-103 benchmark.
>
> |Model|diversity(%)|MAUVE(%)|gen-length|coherence|
> |:-------------:|:-------------:|:-----------------------:|:-----------------------:|:-----------------------:|
> |SimCTG|92.24|85.32|110.34|**-1.43**|
> |Off-the-shelf|**92.48**|**85.66**|108.97|-1.46|
>
> **Table 1: Automatic evaluation results on Wikitext-103.**
>
> |Method A is Better|Neutral|Method B is Better|
> |:-------------:|:-------------:|:-----------------------:|
> |Off-the-shelf **28.2%**|44.9%|26.9% SimCTG|
>
> **Table 2: Human evaluation results on Wikitext-103.**
>
> We evaluate the two compared models through automatic and human evaluations. The evaluated results are presented in Table 1 and 2, respectively. We can see that Off-the-shelf performs comparably with SimCTG on both evaluations. These results further suggest that, when the LM (e.g. GPT-2-large) is intrinsically isotropic, the additional training of SimCTG may not be necessary for contrastive search to work well. On the other hand, when the LM is anisotropic (e.g. GPT-2-small), the training of SimCTG is indispensable as demonstrated by previous work [1].
>
> ### 3. Correlation study between isotropy and variance of degeneration penalty:
> In the Appendix K of our revised manuscript, we evaluate the correlation between isotropy and averaged variance of degeneration penalty using GPT-2, GPT-Neo, and OPT models. Our results in Figure 8 clearly suggest a positive correlation between the LM’s isotropy and the averaged variance of degeneration penalty. These results further validate that a high isotropy of the LM is desirable as it improves the variance of degeneration penalty, therefore benefiting the performance of contrastive search.
>
> ### 4. Other suggested changes on writings:
> Thank you for your suggestions on the writings of our work. We have revised our manuscript corresponding to your suggestions as below:
> * We have revised the description of degeneration penalty in section 2.2. to make it clearer.
> * We have fixed the typo in the size of GPT-2-large in section 3.1.
> * The points in section 4.1.1. have been merged together to make the writings clearer.
>
> ### Reference:
> > [1] Su et al. 2022 "A Contrastive Framework for Neural Text Generation"

---

> > ### Comment · Reviewer_BDTk · 2023-01-14
> > **Thanks for the author response and updated paper.**
> >
> > Thanks very much. I have reviewed the revised manuscript and can confirm that all of the concerns I raised in my review have been addressed.
> >
> > I really appreciate the two new experiments in the appendix. I think Appendix L especially helps solidify the core argument of the paper. The fact that Appendix L includes both automatic and human evaluation is a very nice touch.
> >
> > I have also reviewed the other reviews (and their responses), and think I can safely say I have no further concerns at this point.

---

> > > ### Author Response · Authors · 2023-01-14
> > > **Many thanks for reading our response!**
> > >
> > > Dear Reviewer BDTk,
> > >
> > > Thank you very much for reading our response!
> > >
> > > The Authors

---

### Review · Reviewer_d9f1 · 2023-01-06

**Summary Of Contributions:**

In this paper, the authors conduct extensive empirical studies on pre-trained larger scale autoregressive language models, with two major conclusions. Firstly, different from previous works, the authors find that most LMs are isotropic. Secondly, decoding with contrastive search is better than existing decoding algorithms over different languages and downstream tasks.

**Audience:**

Yes

**Claims And Evidence:**

Yes

**Requested Changes:**

1. Try to make contrastive search also works for anisotropic LMs, or other contributions that increase the impact of the paper.
2. Try different evaluation metrics for MT.

**Strengths And Weaknesses:**

Strengths:
1. The authors conduct extensive experiments and empirical explorations, which is very helpful for the research on this field.
2. The paper is generally well-written and easy to follow.

Weaknesses:
1. The contribution is incremental. The experimental conclusions are somewhat trivial. For example, as an advancing method, it is not surprising that decoding with contrastive search achieves better performance than traditional decoding methods, and previous works have also proved this.
2. The connection between Section 3 and 4 is weak. Given the findings that most LMs are isotropic, is it possible to improve the performance of contrastive search? In Section 8 the authors find that isotropic is essential for contrastive search, then a natural question is, how to make contrastive search also works for anisotropic LMs?
3. It seems that the conclusion that contrastive search is better does not hold for MT when evaluated with BLEU. I recommend to try semantic metrics such as COMET to see whether the gap between contrastive and beam search will be smaller.

---

> ### Author Response · Authors · 2023-01-06
> **Response to the Reviewer d9f1**
>
> Thank you for your thoughtful reviews and suggestions!
>
> ### 1. Make contrastive search work on anisotropic LMs:
>
> We would like to point out that this problem has been investigated in [1]. Specifically, to make contrastive search work on anisotropic LMs, a contrastive training objective, i.e. SimCTG [1], should be first applied. After calibrating the LM with SimCTG, an isotropic representation space can be obtained, and then contrastive search can be used.
>
> Moreover, we would like to highlight the significances of our contributions which are universally acknowledged by all other reviewers (Reviewer BDTk and Reviewer XsZM).
>
> * (1) The anisotropy of autoregressive LMs [2] has been studied and accepted by the research community for the past few years. However, our work is the first study that demonstrates that (most) autoregressive LMs **are** isotropic by their nature. And our finding is surprising and worth to be seen by the research community.
> * (2) Based on our first contribution, we further test the previously proposed contrastive search method using **off-the-shelf** LMs on a wide range of tasks. Extensive experimental results show that, without any additional training, contrastive search can work extremely well on **existing** LMs. As pointed out by the Reviewer BDTk, we believe these results can further reinforce our first contribution and will also help to expand the adoption of contrastive search by the community.
>
>
> ### 2. Different evaluation metrics for MT:
>
> In Section 7, i.e. Table 7, of our revised manuscript, we added the evaluation results on MT using the COMET metric. From the results of COMET, we can draw a similar conclusion to the one obtained with the BLEU metric. In other words, beam search outperforms contrastive search on a few evaluations. However, it is worth noting that the performance gap between beam search and contrastive search is smaller on the COMET metric. This indicates that the generated outputs from both methods are semantically more comparable to each other.
>
>
> ### References:
> > [1] Su et al. 2022 "A Contrastive Framework for Neural Text Generation"
>
> > [2] Ethayarajh, 2019 "How Contextual are Contextualized Word Representations? Comparing the Geometry of BERT, ELMo, and GPT-2 Embeddings"

---

### Author Response · Authors · 2023-02-14
**Thank You**

Dear Action Editor and Reviewers,

We have uploaded the camera-ready version of the paper.

We would also like to express our sincere gratitude to you for your continued feedback and engagement with the paper, which helps us to improve our paper in a substantial way.

The Authors

---

### Decision · Action_Editors · 2023-02-06

**Recommendation:** Accept as is

**Comment:**

This paper has two contributions. The first contribution, as one reviewer commented: "helps avoid the field continuing to advance under false assumptions", is of good value. The second contribution will help expand the adoption of this previously-proposed method. The authors respond actively to address the concerns raised by the reviewer, which are well appreciated. Overall, it is a good paper and I recommend to accept.

**Audience:**

The findings in this work will be of interest to readers who work in text generation.

**Claims And Evidence:**

This paper first points out that most previous language models are isotropic and further demonstrates that decoding with contrastive search is better than existing decoding algorithms over different languages and downstream tasks. The experiments and analysis support the claims well. Although the reviewers propose some problems, the authors actively respond and resolve the weakness.